# Exploring the Antimelanoma Potential of Betulinic Acid Esters and Their Liposomal Nanoformulations

Andreea Milan [1,2], Marius Mioc [1,2], Alexandra Mioc [2,3,*], Narcisa Marangoci [4], Roxana Racoviceanu [1,2], Gabriel Mardale [2,3], Mihaela Bălan-Porcărașu [4], Slavița Rotunjanu [2,3], Irina Șoica [5] and Codruța Șoica [2,3]

1 Department of Pharmaceutical Chemistry, Faculty of Pharmacy, Victor Babes University of Medicine and Pharmacy, Eftimie Murgu Square, No. 2, 300041 Timișoara, Romania; andreea.milan@umft.ro (A.M.); marius.mioc@umft.ro (M.M.); babuta.roxana@umft.ro (R.R.)
2 Research Centre for Pharmaco-Toxicological Evaluation, Victor Babes University of Medicine and Pharmacy, Eftimie Murgu Square, No. 2, 300041 Timișoara, Romania; mardale.gabriel@umft.ro (G.M.); slavita.rotunjanu@umft.ro (S.R.); codrutasoica@umft.ro (C.Ș.)
3 Department of Pharmacology-Pharmacotherapy, Faculty of Pharmacy, Victor Babes University of Medicine and Pharmacy, Eftimie Murgu Square, No. 2, 300041 Timișoara, Romania
4 Institute of Macromolecular Chemistry 'Petru Poni', 700487 Iasi, Romania; nmarangoci@icmpp.ro (N.M.); mihaela.balan@icmpp.ro (M.B.-P.)
5 University College London Medical School, 74 Huntley St., London WC1E 6DE, UK; irina.soica.20@ucl.ac.uk
* Correspondence: alexandra.mioc@umft.ro

**Abstract:** Betulinic acid is a naturally occurring pentacyclic triterpene belonging to the lupane-group that exhibits a wide range of pharmacological activities. BA derivatives are continuously being researched due to their improved anticancer efficacy and bioavailability. The current research was conducted in order to determine the antiproliferative potential of three synthesized BA fatty esters using palmitic, stearic and butyric acids and their liposomal nanoformulations. The cytotoxic potential of BA fatty esters (Pal-BA, St-BA, But-BA) and their respective liposomal formulations (Pal-BA-Lip, St-BA-Lip, But-BA-Lip) has been assessed on HaCaT immortalized human keratinocytes and A375 human melanoma cells. Both the esters and their liposomes acted as cytotoxic agents against melanoma cells in a time- and dose-dependent manner. The butyryl ester But-BA outperformed BA in terms of cytotoxicity (IC$_{50}$ 60.77 μM) while the nanoformulations St-BA-Lip, But-BA-Lip and BA-Lip also displayed IC$_{50}$ values (60.11, 50.71 and 59.01 μM) lower compared to BA (IC$_{50}$ 65.9 μM). The morphological evaluation revealed that the A375 cells underwent morphological changes consistent with apoptosis following 48 h treatment with the tested compounds, while the HaCaT cells' morphology remained unaltered. Both the esters and their liposomal formulations were able to inhibit the migration of the melanoma cells, suggesting a significant antimetastatic effect. The quantitative real-time PCR revealed that all tested samples were able to significantly increase the expression of the pro-apoptotic Bax and inhibit the anti-apoptotic Bcl-2 proteins. This effect was more potent in the case of liposomal nanoformulations versus non-encapsulated compounds, and overall, But-BA and its formulation exhibited the best results in this regard.

**Keywords:** betulinic acid; betulinic acid derivatives; liposomal formulation; cytotoxicity; melanoma

## 1. Introduction

Natural products have long been a major focus in finding treatments for a wide range of maladies. Their enormous pharmacological potential and formulation versatility have designated them as an interesting starting point in the development of different drugs [1]. Several breakthroughs in organic chemistry have been inspired by natural compounds, leading to significantly improved semisynthetic analogues that retain the main scaffold of the natural compound but exert enhanced pharmacological properties [2].

Betulinic acid (BA, 3β-hydroxy-lup-20(29)-en-28-oic acid) is a pentacyclic triterpene, belonging to the lupane group and widely distributed throughout the plant kingdom, but

mainly found in the birch tree bark (*Betula* sp., Betulaceae) [3]. Due to Pisha's discovery that betulinic acid (BA) exhibits specific cytotoxic effects against human melanoma, the pentacyclic triterpene has been analyzed and further developed for its various biological properties [4]. BA exerts significant in vitro cytotoxic effects against a plethora of tumor cell lines, its properties being demonstrated against colon, breast, prostate, hepatocellular, bladder, neck, pancreatic, lung, ovarian and human melanoma [5]. Moreover, several researches have been conducted to confirm and identify BA's mechanism of action as an anticancer, anti-inflammatory, antioxidant, antidiabetic, antiviral, cardioprotective, neuroprotective agent [6–8]. Despite its high pharmacological potential, its current use in therapy is limited particularly by its low in vivo bioavailability [9], a challenge that was tackled both technically through cyclodextrin complexation [10] and liposomal nanoformulations [11,12], and by chemical modulation [13]. Amongst different chemical derivatizations, the esterification with fatty acids has emerged as a promising method for the synthesis of active compounds with improved biological activities. Fatty acids have been identified as cell apoptosis inducers as well as inhibitors of cancer cell proliferation [14]. Al-Hwaiti et al. have demonstrated that palmitic acid and stearic acid, alongside oleic and linoleic acids exerted anticancer effects against colorectal cancer Caco-2 and HCT-116 cells [15]; furthermore, ω-hydroxypalmitic acid and ω-hydroxystearic acid were able to induce cell apoptosis against G361 melanoma cells [16]. It was also reported that butyric acid could facilitate the chemoprevention in colorectal carcinogenesis; Chodurek et al. have tested its chemopreventive effect against A375 melanoma cells while also revealing that sodium butyrate was able to inhibit cell proliferation [17]. Long-chain BA and betulin fatty esters have been previously synthesized by Pinzaru et al. who assessed their anticancer activity, revealing improved pharmacological potential compared to the parent active compound [18]. Furthermore, the evaluation of other pentacyclic triterpenes fatty esters has been performed by Mallavadhani et al. who had synthesized 3-O-fatty ester chains ($C_{12}$-$C_{18}$) of amyrins and ursolic acid, the dodecanoate derivatives showing potent antimicrobial activity against the Gram—*P. syringae*, significantly higher compared to the reference tetracycline [19]. Pentacyclic triterpenes' antiproteolytic effectiveness of the anti-inflammatory potential has been evaluated by Hodges et al.; the authors have obtained two fatty acid esters analogues of lupeol using palmitic and linoleic acids and demonstrated their selective trypsin inhibition properties [20]. Similarly, the synthesis of fatty acids ester derivatives of lupeol by Fotie et al. proved that the introduction of the long side chain has a positive effect on the antimalarial activity against drug-resistant clones of *Plasmodium falciparum* W-2 and D-6 [21].

To address the bioavailability issue associated with the highly lipophilic nature of BA fatty esters, their inclusion in liposomal formulations has been regarded as a potentially viable solution. Due to their unique chemical and physical features such as their amphiphilic characteristics, resemblance to human cells and the possibility of being extensively surface-modified, liposomes provide a myriad of advantages over other nanoparticles, including great preparation versatility, the capacity to encapsulate a large number of distinct compounds, and targeted delivery resulting in high patient tolerance [22]. However, conventional liposomes are easily unstable in the plasma due to their chemical composition and their interaction with lipoproteins [23]; sterically stabilized through surface modifications (stealth) liposomes are able to prevent drug leakage prior to delivery [24].

The current study proposes the synthesis of BA fatty esters using stearic, palmitic and butyric acids, which are reported to possess individually unique pharmacological effects [25–27] followed by their encapsulation in surface-modified liposomal nanoformulations bearing polyethylene glycol fragments in order to prolong their lifespan. The resulting formulations were assessed as anticancer agents against human malignant melanoma (A375) cells.

## 2. Results

*2.1. Chemistry*

2.1.1. Synthesis and Characterization of Fatty Acid Esters of BA

The reaction conditions for obtaining fatty BA esters are depicted in Figure 1. All three compounds were obtained in high yields (>65%). The $^1$H NMR spectra of the esters' derivatives show the peaks for the two H30 protons from the BA backbone as two singlets at 4.74 and 4.61 ppm, H3 resonates at 4.47 ppm, H19 is found at 3.00 ppm and the $CH_2$ protons adjacent to the ester group of the fatty acid residue resonate at 2.30 and 2.27 ppm, overlapped with one of the H15 protons from the BA. The integral values of the peaks from the $^1$H NMR spectra are in accordance with the proposed structures (Figures S1, S7 and S13, Supplementary Materials). The $^{13}$C NMR spectra show the peak for the COOH group from BA at around 182 ppm, the COO carbon of the ester group at about 173 ppm and the peaks for the rest of the carbon atoms at the appropriate chemical shift values (Figures S2, S8 and S14, Supplementary Materials). The peak for the ester carbon from 173 ppm gives long range correlation peaks in the H,C-HMBC spectra (Figures S6, S12 and S18, Supplementary Materials) with both H3 from BA and with $CH_2$ from the fatty acid chain, thus demonstrating the covalent bonding between BA and the fatty acids. Ester formation is also supported by FTIR spectroscopy, which shows an adjacent C=O signal belonging to the ester function at 1730 cm$^{-1}$ in the spectra of each compound. Furthermore, the strong OH signal from the BA spectra at 3349 cm$^{-1}$ is no longer present in the FTIR spectra of the obtained esters (Figure S19, Supplementary Materials). The most relevant physicochemical properties of each newly synthetized BA derivative are presented in Table 1.

**Figure 1.** Synthesis of BA fatty acid ester derivatives; BA: betulinic acid, But-BA: 3-O-butiryl-betulinic acid, Pal-BA: 3-O-palmitoyl-betulinic acid, St-BA: 3-O-stearoyl-betulinic acid; DCM: dichloromethane, DMAP: 4-Dimethylaminopyridine.

**Table 1.** Physicochemical properties of synthetized BA derivatives.

| | Melting Point | Yield | Appearance | *m/z* [M-H+]$^-$ |
|---|---|---|---|---|
| Compound | | | | |
| But-BA | 265–280 °C | 74% | white powder | 525 |
| Pal-BA | 160–170 °C | 70% | translucent crystals | 693 |
| St-BA | 150–162 °C | 65% | translucent crystals | 721 |

2.1.2. Synthesis and Characterization of BA-Fatty Acid Ester Liposomal Formulation

The lipid film hydration method, which has previously been shown to be suitable for our current purpose [28], was used to obtain bare liposomes as well as liposomal formulations loaded with BA and its newly synthesized esters. Transmission electron microscopy (TEM), scanning electron microscopy (SEM), and dynamic light scattering (DLS) were used to examine the liposomes. Results are depicted in Figures 2–4. The TEM

and SEM images revealed stable spherical liposomes of various sizes. The diameters of the bare liposomes were rarely larger than 100 nm, whereas the addition of triterpenes increased their diameter. The liposomal formulations containing the palmitoyl and stearoyl esters of BA displayed the largest particles, with sizes occasionally exceeding 200 nm (Figures 2 and 3). While the microscopy-recorded diameters and the mean hydrodynamic size of the particles slightly differ, DLS measurements were within the expected range (Figure 4). The measured polydispersity index was roughly related to the measured liposome diameters; bare liposomes had the lowest dispersity (PI 0.1955), BA-Lip and But-BA-Lip had PI values in the around 0.2, while the large ester formulations exhibited PI values in the 0.4–0.5 range (Figure 4). DLS measurements were repeated daily for one week; no significant changes in the recorded PI values and hydrodynamic size occurred, indicating that these formulations were stable within the tested time period. The determined $\zeta$ potential values for empty liposomes, BA-Lip, But-BA-Lip, Pal-BA-Lip and St-BA-Lip were $-22.1$ mV, $-19.4$ mV, $-18.7$ mV, $-18.2$ mV and $-17.8$ mV, respectively. Drug loading efficiency (DLE) ranged from 78 to 85% for all liposomal formulations; DLE values for BA-Lip, But-BA-Lip, Pal-BA-Lip and St-BA-Lip were 78%, 85%, 82% and 80%, respectively. Furthermore, liposome stability related to drug loading efficiency was assessed over a 15-day period at two different temperatures. As depicted in Table 2, minor changes in drug encapsulation efficiency were observed during the stability study at 4 °C, but there was a significant decrease in stability when the formulations were stored at 25 °C. As indicated by the percentage difference between day 1 and day 15, liposomes containing BA were the most stable formulation, while But-BA-Lip was the least stable.

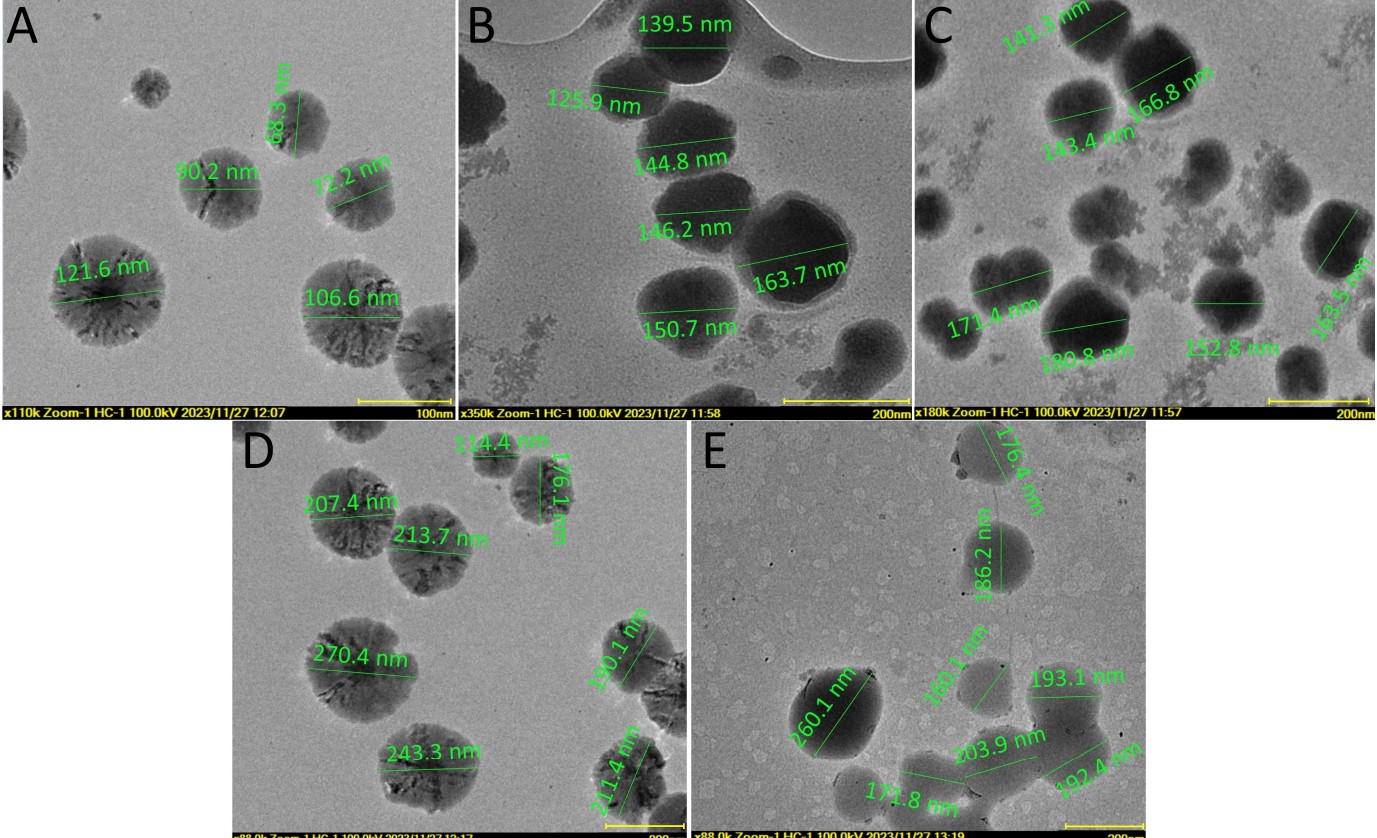

**Figure 2.** TEM images of (**A**) bare liposome (scale bar 100 nm), (**B**) BA-Lip (scale bar 200 nm), (**C**) But-BA-Lip (scale bar 200 nm), (**D**) Pal-BA-Lip (scale bar 200 nm) and (**E**) St-BA-Lip (scale bar 200 nm).

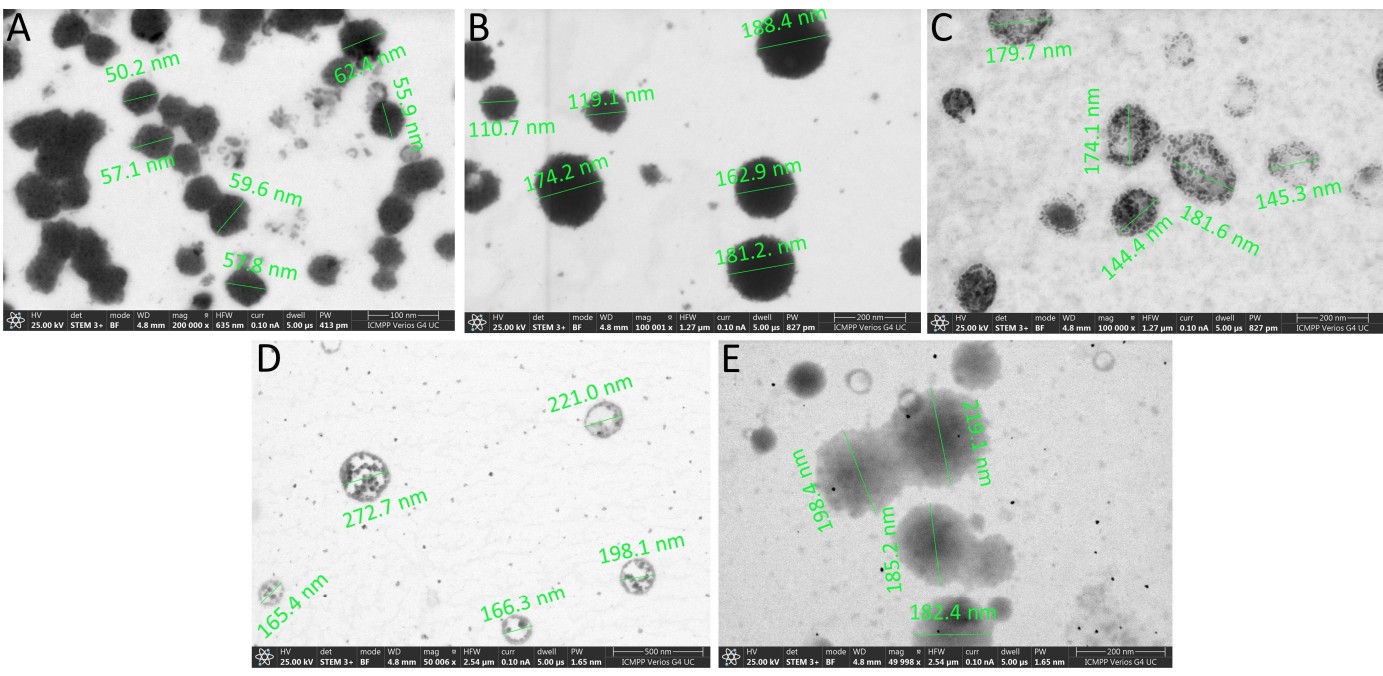

**Figure 3.** SEM images of (**A**) bare liposome (scale bar 100 nm), (**B**) BA-Lip (scale bar 200 nm), (**C**) But-BA-Lip (scale bar 200 nm), (**D**) Pal-BA-Lip (scale bar 500 nm) and (**E**) St-BA-Lip (scale bar 200 nm).

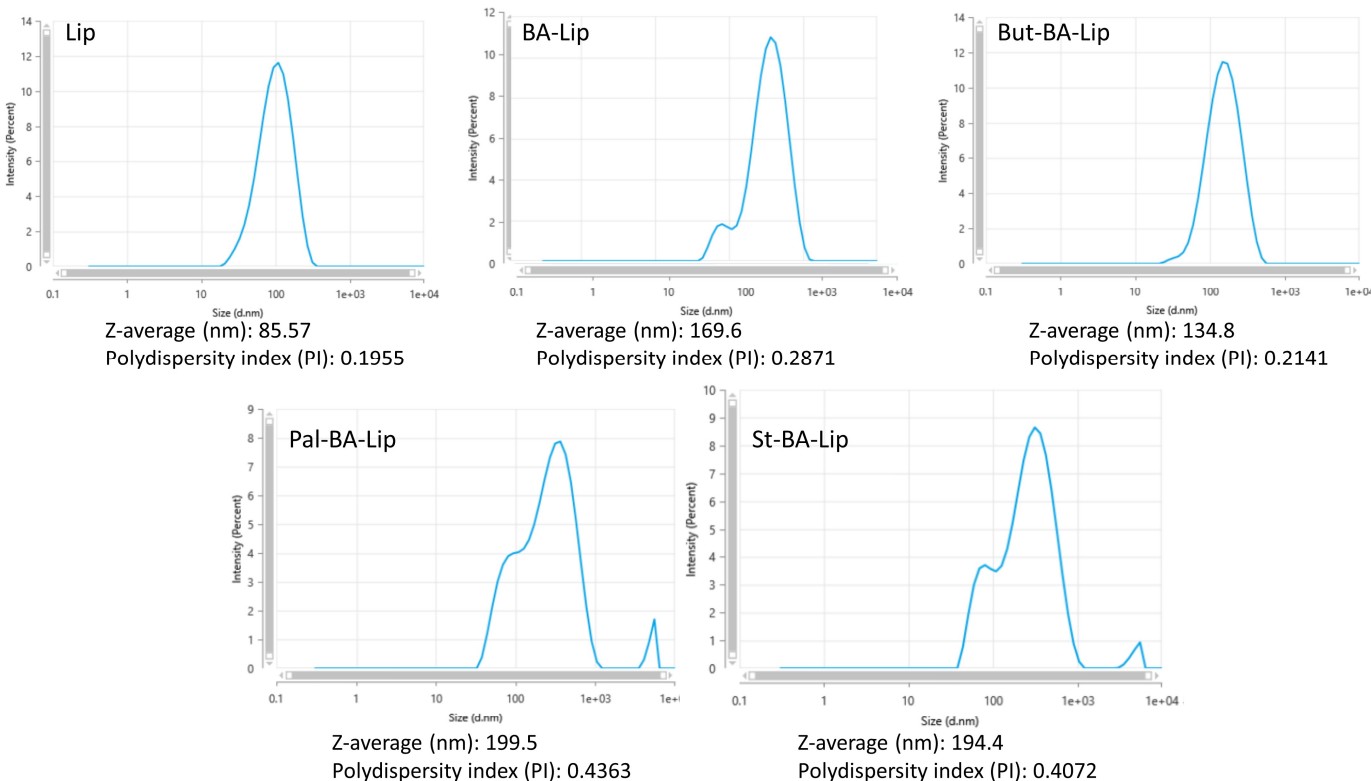

**Figure 4.** Measured hydrodynamic size and polydispersity index of the obtained liposomal formulations.

**Table 2.** Drug loading efficiency of liposomal formulations determined at 4 °C and 25 °C, for 15 days.

| | Drug Loading Efficiency (DLE %) | | | | | | | |
| --- | --- | --- | --- | --- | --- | --- | --- | --- |
| | **BA-Lip** | | **But-BA-Lip** | | **Pal-BA-Lip** | | **St-BA-Lip** | |
| | **4 °C** | **25 °C** | **4 °C** | **25 °C** | **4 °C** | **25 °C** | **4 °C** | **25 °C** |
| **Day 1** | 78.12 | 78.12 | 85.22 | 85.22 | 82.06 | 82.06 | 80.18 | 80.18 |
| **Day 3** | 78.08 | 77.66 | 84.91 | 84.51 | 81.65 | 81.42 | 79.72 | 79.71 |
| **Day 5** | 77.65 | 77.13 | 84.58 | 83.72 | 81.17 | 80.68 | 79.16 | 78.88 |
| **Day 7** | 76.87 | 76.35 | 83.96 | 82.82 | 80.58 | 79.81 | 78.44 | 77.85 |
| **Day 9** | 76.11 | 75.38 | 83.15 | 80.91 | 79.83 | 78.87 | 77.53 | 76.71 |
| **Day 11** | 75.76 | 74.4 | 82.21 | 78.82 | 78.95 | 77.72 | 76.49 | 75.46 |
| **Day 13** | 74.58 | 73.17 | 81.03 | 76.51 | 78.01 | 76.25 | 75.35 | 73.89 |
| **Day 15** | 73.89 | 71.11 | 78.84 | 73.64 | 76.63 | 74.56 | 74.07 | 72.14 |
| **Day 1–15 difference** | 4.23 | 7.01 | 6.38 | 11.58 | 5.43 | 7.5 | 6.11 | 8.04 |

### 2.2. Evaluation of Betulinic Acid Fatty Esters and Liposomes Cytotoxic Effect

The viability of nonmalignant human keratinocytes—HaCaT and human malignant melanoma—A375 cells was evaluated at 24 h and 48 h post-treatment with the newly synthesized compounds (10, 25, 50, 75 and 100 μM) using the Alamar blue assay. The incubation of nonmalignant HaCaT cells for 24 and 48 h revealed that, except for But-BA and But-BA-Lip, the tested compounds did not exhibit cytotoxic effects against HaCaT cells even at the highest tested concentrations (75 and 100 μM). However, the slightly cytotoxic effects of But-BA and But-BA-Lip were only recorded at the highest tested concentration with cell viability (%) decreasing at 73.15 ± 3.9 (48 h), 75.24 ± 2.57 (24 h) for But-BA and 84.12 ± 1.5 (48 h), 84.92 ± 0.06 (24 h) for But-BA-Lip; neither effect was comparable to 5-FU (5-Fluorouracil) where cell viabilities dropped to 30.61 ± 3.54 (48 h), 35.58 ± 2.87 (24 h) when the same concentration (100 μM) was applied (Figure 5).

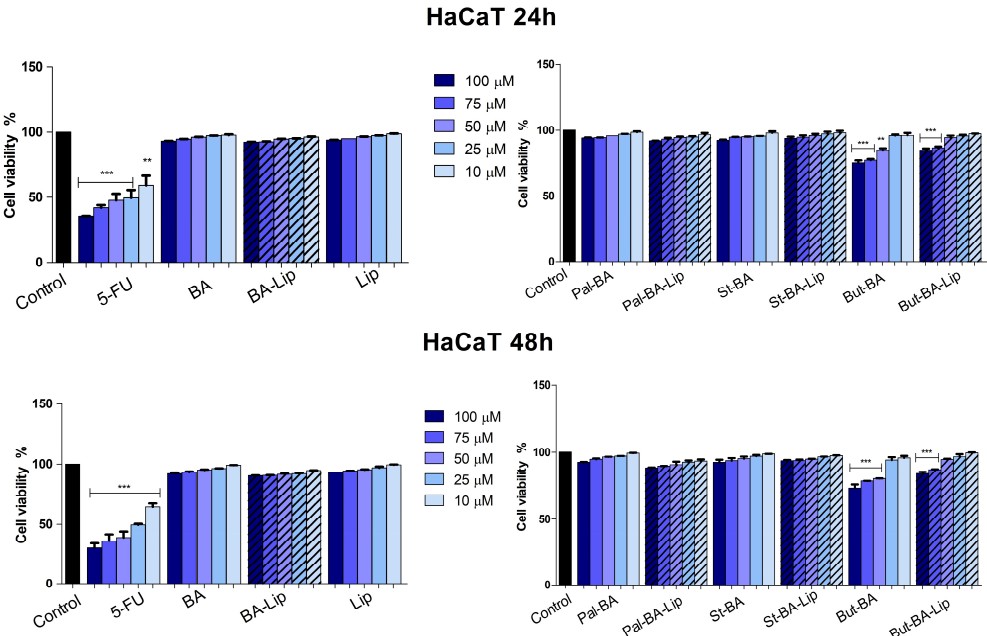

**Figure 5.** Cell viability after 24 and 48 h treatment with 5-FU, BA, BA-Lip, Pal-BA, Pal-BA-Lip, St-BA, St-BA-Lip, But-BA and But-BA-Lip (10, 25, 50, 75 and 100 μM) on HaCaT cells. The results are expressed as viability percentages compared to the control group, considered 100% (** *p* < 0.01 and *** *p* < 0.001 vs. control cells). The slash bars are represented by the liposomal nanoformulations of BA's esters. The data represents the mean values ± SD of three independent experiments performed in triplicate.

In terms of antimelanoma effects, the results revealed that among the synthetized esters, only But-BA decreased cell viability more aggressively than BA alone subsequently displaying lower $IC_{50}$ value after 48 h incubation (60.77 μM vs. 65.9 μM) (Table 3); the other esters (Pal-BA and St-BA) significantly decreased cell viability (%) vs. control (100%) but displayed a similar cytotoxic profile with pure BA when the highest concentrations were applied for 48 h (47.29 ± 1.39 for Pal-BA and 45.54 ± 2.8 for St-BA vs. 40.24 ± 0.89 for BA). Moreover, the results showed that the inclusion of BA or its fatty esters in liposomes induced stronger cytotoxic effects compared to the fatty esters and pure BA, respectively; however, neither compound was able to match the antiproliferative activity of 5-FU, as follows: 37.34 ± 2.74 (Pal-BA-Lip 100 μM–48 h), 46.73 ± 4.96 (Pal-BA-Lip 100 μM–24 h), 44.52 ± 0.52 (St-BA-Lip 100 μM–48 h), 45.43 ± 1.68 (St-BA-Lip 100 μM–24 h), 33.42 ± 0.03 (But-BA-Lip 100 μM–48 h), 42.94 ± 4.87 (But-BA-Lip 100 μM–24 h), 40.24 ± 0.89 (BA 100 μM–48 h) and 52.47 ± 1.01 (BA 100 μM–24 h) compared to 20.23 ± 2.24 (5-FU 100 μM, 48 h) and 26.45 ± 7.53 (5-FU 100 μM–24 h). Empty liposomes did not exert cytotoxic effects in either HaCaT human keratinocytes or melanoma A375 cells (Figure 6).

**Table 3.** The calculated $IC_{50}$ values (μM) of 5-FU, BA, BA-Lip, Pal-BA, Pal-BA-Lip, St-BA, St-BA-Lip, But-BA and But-BA-Lip on HaCaT and A375 cell lines 48 h post-stimulation.

| Compounds | HaCaT | A375 |
|---|---|---|
| 5-FU | 40.14 ± 1.2 | 26.61 ± 0.82 |
| BA | >100 | 65.9 ± 1.07 |
| BA-LIP | >100 | 59.01 ± 0.45 |
| PAL-BA | >100 | 85.58 ± 1.32 |
| PAL-BA-LIP | >100 | 67.59 ± 0.33 |
| ST-BA | >100 | 75.75 ± 0.75 |
| ST-BA-LIP | >100 | 60.11 ±1.56 |
| BUT-BA | >100 | 60.77 ± 0.29 |
| BUT-BA-LIP | >100 | 50.71 ± 0.67 |

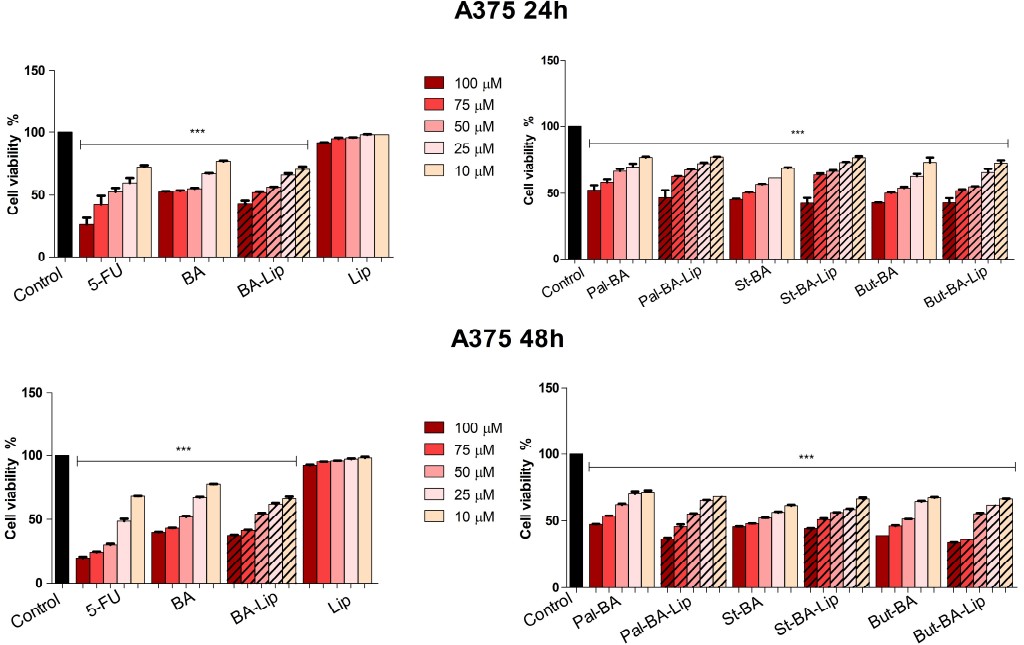

**Figure 6.** Cell viability after 24 and 48 h treatment with 5-FU, BA, BA-Lip, Pal-BA, Pal-BA-Lip, St-BA, St-BA-Lip, But-BA and But-BA-Lip (10, 25, 50, 75 and 100 μM) on A375 cells. The results are expressed as viability percentages compared to the control group, considered 100% (*** $p < 0.001$ vs. control cells). The slash bars represent viability data for liposomal nanoformulations of BA's esters. The data represents the mean values ± SD of three independent experiments performed in triplicate.

### 2.3. Fatty Ester Derivatives Effects on Cell Morphology

In nonmalignant HaCaT cells, no significant differences have been observed in terms of cell morphology and confluence between treated and untreated (control) cells after 48 h. But-BA (100 μM) and But-BA-Lip (100 μM) only slightly altered cellular morphology, some of the cells becoming rounder and on the verge of detaching (Figure S20A–C, Supplementary Materials). The positive control, 5-FU, decreased the number and altered HaCaT morphology and confluence, making them round and detached (Figure S20A).

In A375 melanoma cells, treatment with 5-FU, the highest concentration of esters (Pal-BA, St-BA and But-BA) and their liposomes (Pal-BA-Lip, St-BA-Lip and But-BA-Lip) induced several morphological changes such as round and detached cells, changes that occurred simultaneously with a reduced number of cells, thus correlating to viability results (Figure S21A–C, Supplementary Materials).

No significant changes in HaCaT cell morphology were detected in terms of cell cytoskeleton architecture and nuclei upon treatment with BA, esters or their liposomal formulations (BA, Ba-Lip, But-BA, But-Ba-Lip, Pal-BA, Pal-BA-Lip, St-BA and St-BA-Lip) in the highest tested concentration (100 μM) for 48 h (Figure 7A,B).

## HaCaT

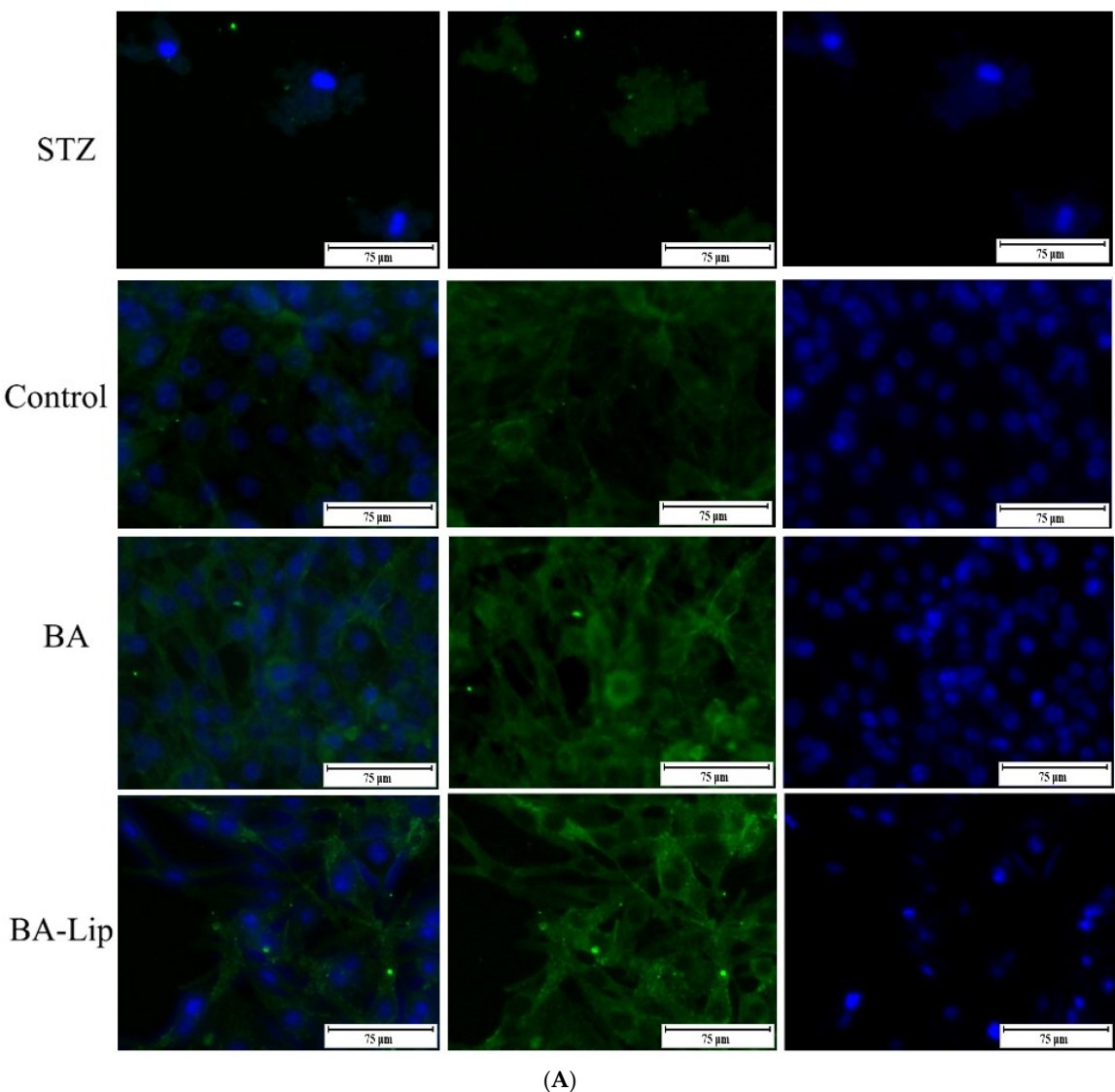

(**A**)

**Figure 7.** *Cont.*

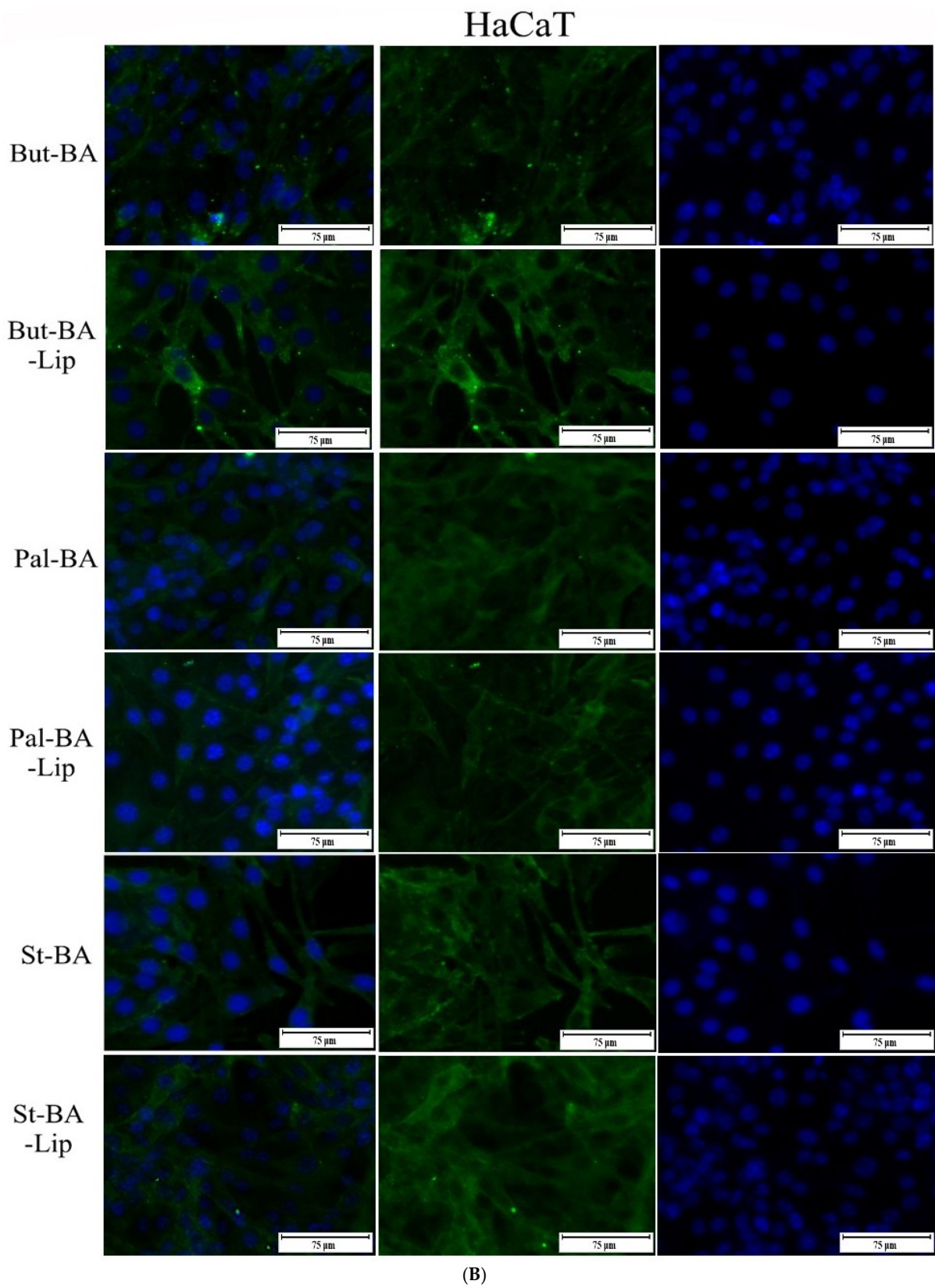

(**B**)

**Figure 7.** The impact of 48 h treatment with BA, Ba-Lip (100 μM) (**A**) and But-BA, But-Ba-Lip, Pal-BA, Pal-BA-Lip, St-BA and St-BA-Lip (100 μM) (**B**) on HaCaT cells nuclei (blue–Hoechst staining–third column), cytoskeleton (beta-actin–green staining–second column) and the merged picture (first column). Staurosporine (STZ, 5 μM) was used as a positive control for necrotic cell death. The scale bar was 150 μm.

The 48 h treatment of A375 melanoma cells with BA, fatty acid esters and their liposomal formulation (Ba-Lip, But-BA, But-Ba-Lip, Pal-BA, Pal-BA-Lip, St-BA and St-BA-Lip) using their $IC_{50}$ values induced various morphological changes that are consistent with apoptosis (Figure 8A,B). In particular, the treated cells underwent observable cytoskeletal rearrangement accompanied by the alteration of cell shape and loss of structural integrity; other morphological changes can be observed at nuclear level, such as: chromatin condensation (the nucleus appears highly compacted), nuclear shrinkage and fragmentation and the formation of apoptotic bodies that can be seen as widely spread small condensed chromatin fragments with various sizes.

Quantitative real-time PCR was used to determine gene expression variations of anti-apoptotic Bcl-2 and pro-apoptotic Bax in order to further investigate the pro-apoptotic effect of BA, BA-fatty acid esters, and their liposomal formulations against melanoma cells. After a 24-h incubation period, measurements were carried out on A375 cells treated with test compounds at a sub-cytotoxic concentration of 10 μM. The results show that all compounds and formulations increase the expression of the pro-apoptotic Bax gene while decreasing the expression of the anti-apoptotic Bcl-2 gene (Figure 9). BA induced a two-fold reduction in the relative fold expression of Bcl-2. In this case, BA was outperformed by its liposomal formulations Pal-BA and Pal-BA-Lip. This trend could be observed in the case of BAX, where the 2.5 increase in relative fold expression induced by BA was surpassed by the same BA-Lip, Pal-BA, and Pal-BA-Lip. It is also worth noting that the liposomal formulation outperformed the unencapsulated compound in each case.

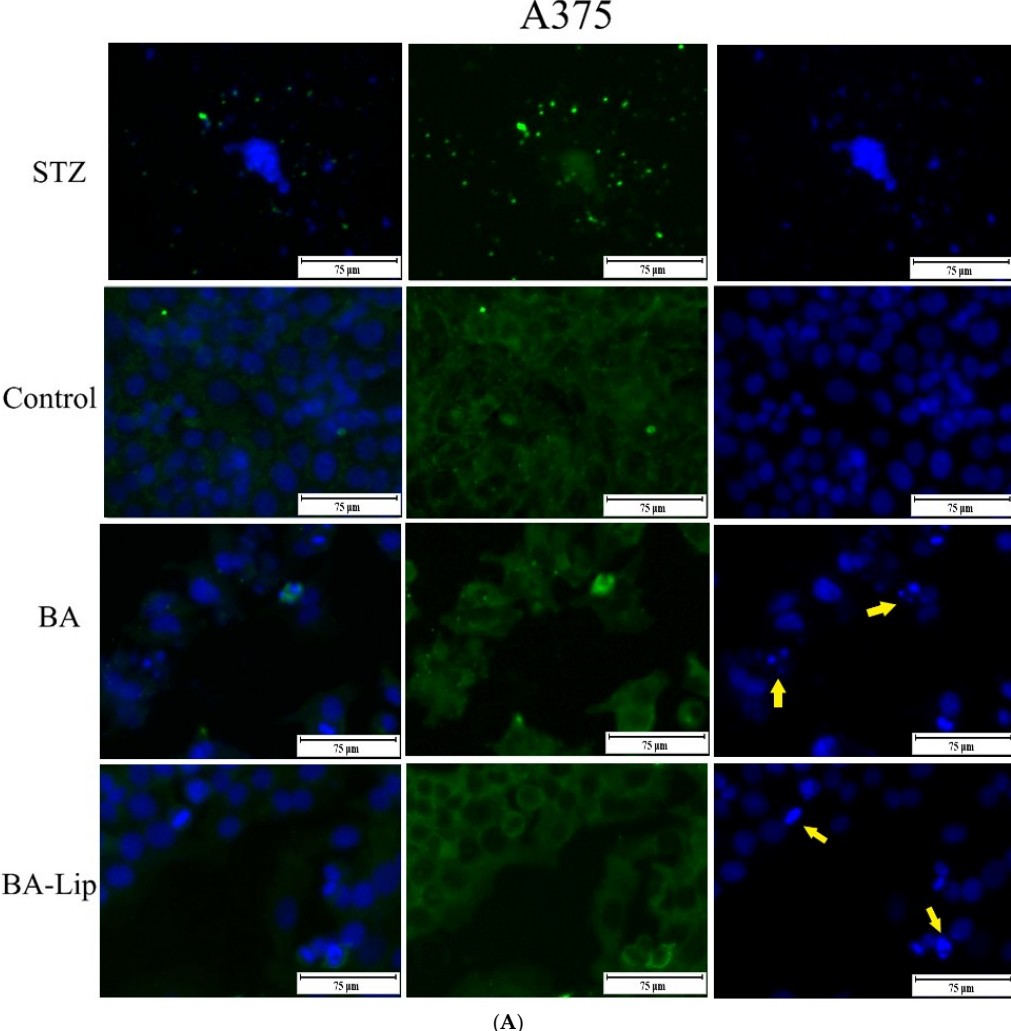

(A)

**Figure 8.** *Cont.*

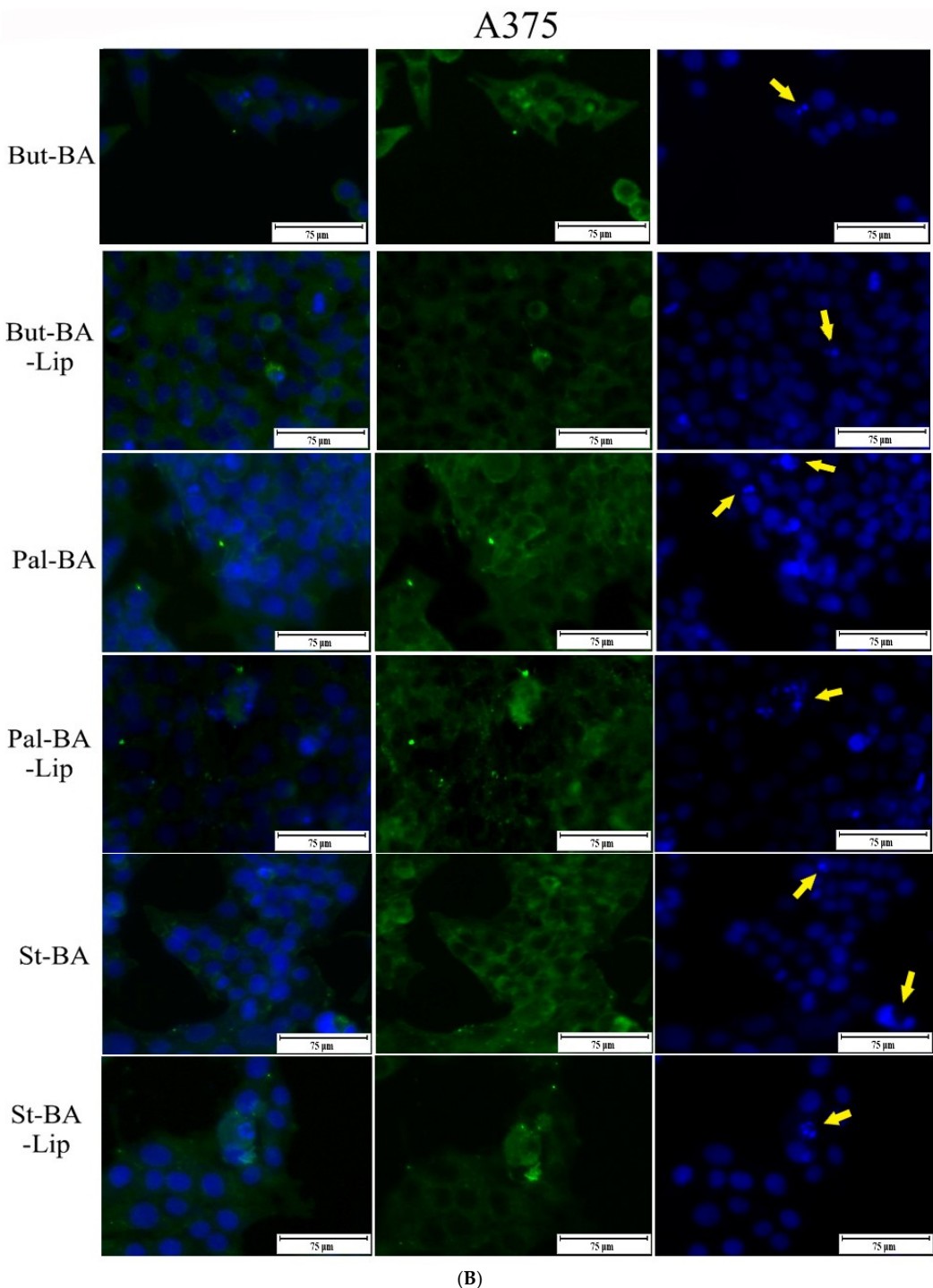

(**B**)

**Figure 8.** The impact of 48 h treatment with BA, Ba-Lip (IC$_{50}$) (**A**) and But-BA, But-Ba-Lip, Pal-BA, Pal-BA-Lip, St-BA and St-BA-Lip (IC$_{50}$) (**B**) on A375 cells nuclei (blue—Hoechst staining—third column), cytoskeleton (beta-actin—green staining—second column) and the merged picture (first column). Staurosporine (STZ, 5 μM) was used as a positive control for necrotic cell death. The yellow arrows indicate signs of apoptotic cell death. The scale bar was 150 μm.

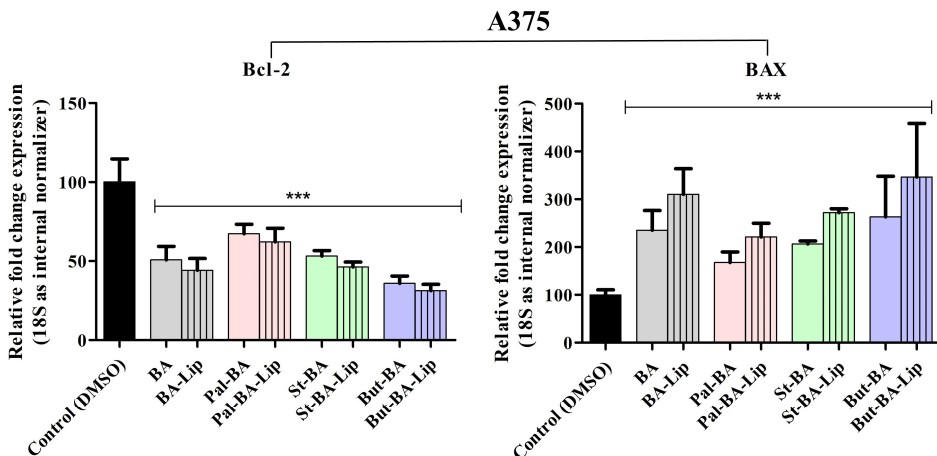

**Figure 9.** Relative fold change expression in mRNA of Bcl-2 and BAX in A375 cells after stimulation with BA, BA-Lip, But-BA, But-Ba-Lip, Pal-BA, Pal-BA-Lip, St-BA and St-BA-Lip at 10 µM. The expressions were normalized to 18S and DMSO was used as control. Data represents the mean values ± SD of three independent experiments. One-way ANOVA with Dunnett's post-test was applied to determine the statistical differences in rapport with DMSO stimulated cells (*** $p < 0.001$ vs. control cells).

*2.4. Scratch Assay*

In order to determine the anti-migratory effects of BA, BA fatty acid esters and their liposomal formulations, a scratch assay technique was performed. Stimulation of A375 melanoma cells with BA esters and their respective liposomes revealed an efficient anti-migratory effect for the tested concentration (10 µM) both 24 h and 48 h post-treatment (Figures 10 and 11A–C). All compounds exhibited scratch closure rates below 30% compared to control with a scratch closure rate of 70% after 24 h and 81.49% after 48 h. BA esters (Pal-BA, St-BA and But-BA) inhibited the scratch closure rate to 14.5%, 23.31% and 19.98%, respectively. The most impressive anti-migratory effects were exhibited by But-BA-Lip, with a scratch closure rate of 8.23%, and Pal-BA-Lip, with a scratch closure rate of 8.61%, values that were slightly lower compared to those of BA at 16.48% and BA-Lip at 9.62%. St-BA-Lip decreased the scratch closure rate to 22.36%. Furthermore, at this concentration, cells with round shape and detached cells could be observed, thus clearly showing the cytotoxic effects of these compounds against melanoma cells.

**A375**

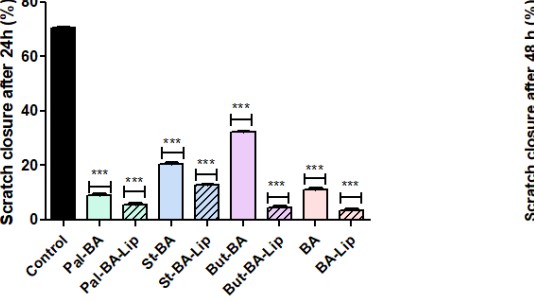
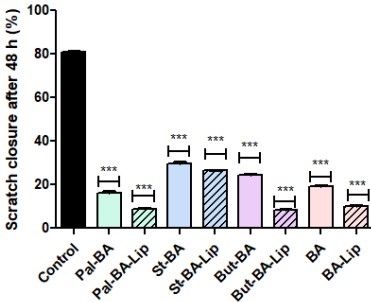

**Figure 10.** Scratch migration assay of Pal-BA, Pal-BA-Lip, St-BA, St-BA-Lip, But-BA, But-BA-Lip, BA and BA-Lip (10 µM) on A375 cells. The percentage signifies the remnant gap size 24 h and 48 h after conducting the scratches compared to the initial gap size. Values were expressed as mean ± SD and the asterisk values show significant results of the tested compounds compared to the control group using the one-way ANOVA's test followed by Dunnett's post-test.

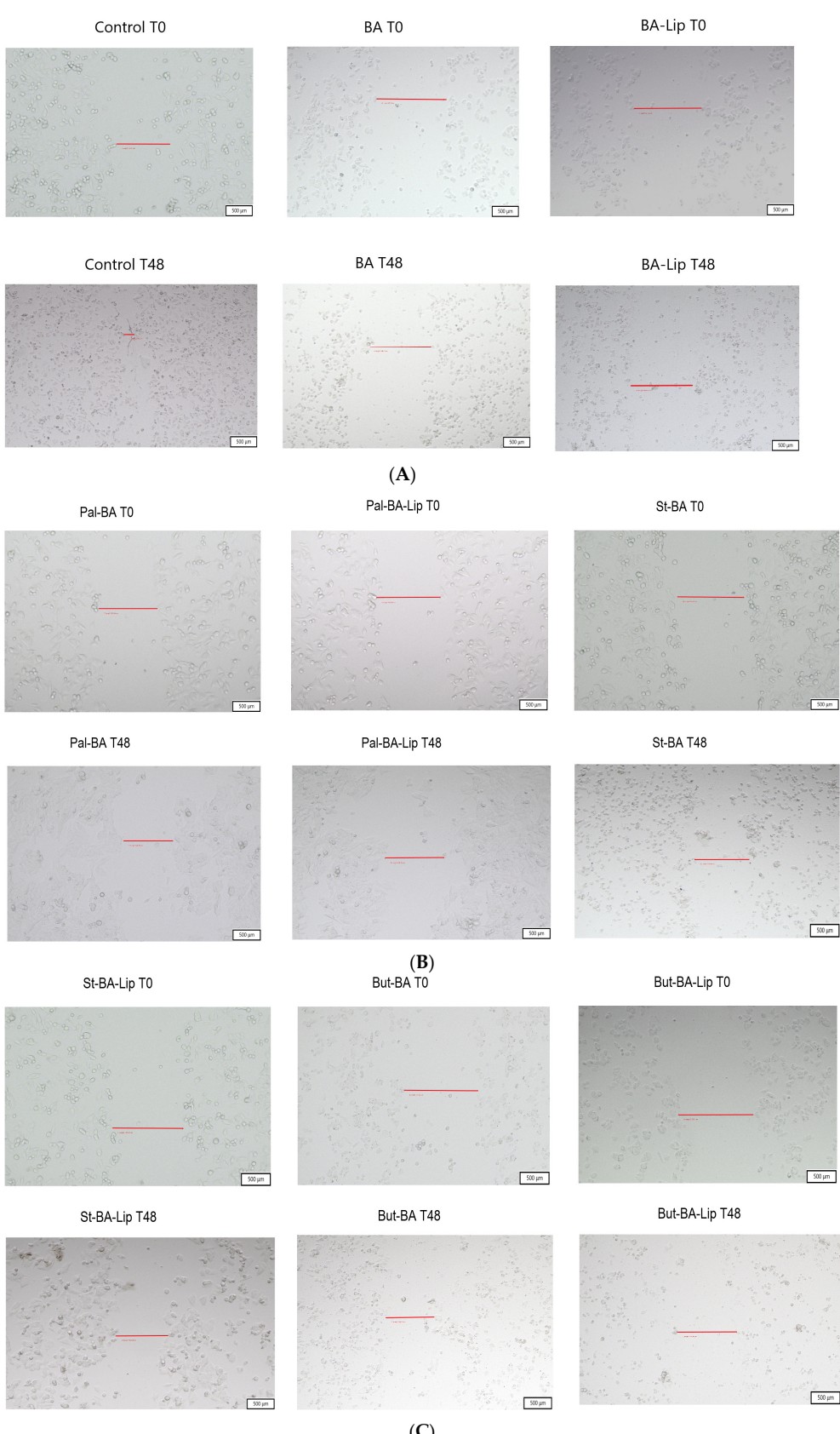

**Figure 11.** The effects of BA and BA-Lip (**A**), Pal-BA, Pal-BA-Lip, St-BA (**B**), St-BA-Lip, But-BA and But-BA-Lip (**C**) (10 µM) on malignant melanoma cells A375 migration capacity. The images were taken by light microscopy at 10× magnification. The cell migration was measured both at 0 h and at 48 h after stimulation. The red line represents the measured width of the scratched/wounded area.

*2.5. Betulinic Acid Esters Effect on BAX and Bcl-2 Protein Levels*

In order to determine whether the treatment with BA, BA-Lip, Pal-BA, Pal-BA-Lip, St-BA, St-BA-Lip, But-BA and But-BA-Lip induce apoptotic cell death in A375 cancer cells, the pro-apoptotic BAX and anti-apoptotic Bcl-2 levels were quantitatively measured using an in vitro Enzyme-Linked Immunosorbent assay (ELISA). The results indicate that all compounds were able to increase the pro-apoptotic BAX protein level, while decreasing the anti-apoptotic Bcl-2 protein level (Figure 12).

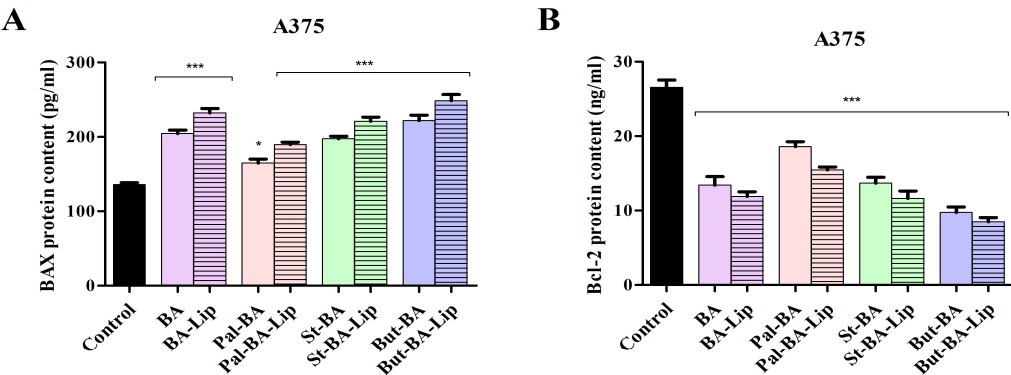

**Figure 12.** Effect of BA, BA-Lip, Pal-BA, Pal-BA-Lip, St-BA, St-BA-Lip, But-BA and But-BA-Lip, (IC50) on BAX (**A**) and Bcl-2 (**B**) protein levels in A375 cell lines after 48 h treatment. The results were reported as mean values $\pm$ SD with $p < 0.05$ (*) and $p < 0.001$ (***), when compared to control. All experiments were performed in triplicate.

## 3. Discussion

When comparing natural compounds versus synthetic ones as treatments against malignant diseases, nature wins points by presenting certain advantages such as superior efficacy and safety, lower costs and chemical diversity; moreover, natural compounds have the ability to modulate multiple oncogenic signaling pathways while conventional chemotherapeutic drugs usually aim at only one particular target [29]. Betulinic acid, which can be found in many tree and plant species, shows a wide range of biologic activities [30] including selective anticancer effects against numerous types of cancer cells [31]; its ability to directly target mitochondria and trigger cancer death provides the possibility of alternative anticancer treatment when conventional therapy fails [32]. Despite its promising anticarcinogenic effects, BA's low bioavailability has impelled many researchers to synthesize new analogues that exhibit improved pharmacokinetic and pharmacodynamic profiles [33]. The chemical modulation of betulinic acid may lead to the synthesis of hybrid molecules that occupy a special place in the development of effective anticancer agents, since they provide the possibility to enhance and enlarge the biological effects of the parent molecule while circumventing drug resistance [34]. Some researchers reported that the biological activities of several pentacyclic triterpenes such as amyrin, oleanolic acid, lupeol and ursolic acid might be improved through esterification with fatty acids [19,35]. Despite general knowledge that saturated fatty acids are associated with increased risks of cardiovascular events, they are also involved in important physiological regulatory mechanisms in protein activation and subcellular trafficking as well as gene transcription; moreover, saturated fatty acids may induce apoptosis through several pathways [36]. To the best of our knowledge, fatty acid conjugates of anticancer agents are rather poorly explored although several studied revealed that both saturated and unsaturated fatty acids have the ability to improve the anticancer efficacy and selectivity of conjugated drugs [37]. Betulinic acid is currently regarded as unsuitable for therapeutic use due to its low solubility and bioavailability; the introduction of a hydrophobic fragment in the molecule's scaffold causes an increase in lipophilicity that may induce an increased uptake of the compound through the cell membrane, thus optimizing its bioavailability [38].

Triterpene esterification can be accomplished using a variety of reagents such as free acids, anhydrides or acyl chlorides; since BA contains a single hydroxyl group that can be acylated, the more reactive acyl chlorides were chosen as reaction partners. One previous study used anhydrides to synthesize various 3-O-BA esters [39], including 3-O-butiryl-BA; however, the resulting compound was obtained in lower yields compared to the current study. BA esters were subsequently incorporated in liposomes which are able to prolong circulation time and facilitate intracellular absorption; in addition, liposome entrapment may delay ester hydrolysis, thus providing in vivo controlled drug release [40].

Several previous studies reported BA-containing liposomes with similar characteristics which were also prepared by using the film hydration method employed in the current study, thus making it highly reproducible. As an example, Farcas et al. described a BA formulation containing magnetic nanoparticles entrapped in PEGylated liposomes [28] where, similar to our case, the addition of BA resulted in an increase in hydrodynamic size and PI values; the authors also reported comparable DLE values. Liu Y et al. prepared BA PEGylated liposomes with a mean diameter of 142 nm and DLE of up to 95% [12]; Mullauer et al. used the film hydration method to obtain BA-liposomes with sizes ranging from 100 to 200 nm [41]. Even when the liposomal formulation contains a mixture of BA with other active compounds, it appears that the DLE reported for BA is maintained; Jin et al. developed a PEGylated liposomal formulation that included BA, parthenolide, honokiol and ginsenoside Rh2 that exhibited 89.5% DLE related to BA alone. To the best of our knowledge, this is the first report on liposomes containing BA esters with fatty acids; their diameters are clearly larger than those recorded for BA formulations presumably due to their higher molecular weight, while their DLE reached similar values (around 80%). For But-BA-Lip, the particle size fell within the range of 100–200 nm, while for the other two liposomal formulations the diameter exceeded 200 nm; however, the tumor neo-angiogenesis process induces the occurrence of gaps between endothelial cells of up to 2 μm that allow a preferential access to tumor sites compared to normal cells that display a tighter structure with 5–10 nm pores [40]. Liposomes remained unchanged in size and PDI values for 7 days. However, when all formulations were tested for 15 days for drug loading fluctuations, we discovered that when stored at room temperature (25 °C), drug loading efficiency decreased significantly, with the But-BA-Lip formulation being the least stable. When the samples were refrigerated (4 °C), the reductions in drug loading efficiency were minor. This phenomenon, in which the drug content of liposomal formulations degrades at higher temperatures during storage, is common in PEGylated liposomes, including those containing pentacyclic triterpenes [12,42]. The observed reduction in $\zeta$ potentials (−22.1 mV, −19.4 mV, −18.7 mV, −18.2 mV, and −17.8 mV) across the five formulations (Lip, Ba-Lip, But-Ba-Lip, Pal-Ba-Lip and St-Ba-Lip) indicates an apparent correlation with the loaded drug's mass. The $\zeta$ potential values slightly decrease as the loaded drug's mass increases. This phenomenon can be caused by the interaction of drug molecules with the liposomes' surface charge. The accumulation of drug molecules towards the liposome's surface may shield the charges, lowering the absolute value of the overall zeta potential. Such fluctuations in zeta potential indicate changes in the electrostatic stability of liposomal formulations [43]. Given that stable liposomes typically have zeta potential values that fall outside of the −30 mV to 30 mV range [44], the observed decrease in zeta potential is not that concerning given that these values fall in the same range (−20 mV–−10 mV) as similar liposomes that were previously reported [28,45,46].

The fatty acids esters (Pal-BA, St-BA, But-BA) as well as their liposomal formulations (Pal-BA-Lip, St-BA-Lip, But-BA-Lip) were assessed against immortalized human keratinocytes HaCaT and human melanoma A375 cells in terms of cell viability using the Alamar blue assay. HaCaT is a non-cancerous monoclonal cell line derived from adult human keratinocytes that can support long-term growth without supplemented growth factors; it displays all morphological features, surface markers and functions of normal keratinocytes and has been used in numerous cell viability studies as model of non-malignant cells for anticancer selectivity assessments [47]. The experimental results revealed that

except for But-BA and But-BA-Lip none of the tested compounds exerted cytotoxic effects against HaCaT keratinocytes, regardless of concentration; these findings indicate that Pal-BA and St-BA, as well as their liposomal formulations, Pal-BA-Lip and St-BA-Lip, may selectively act against cancer cells without affecting healthy cells even when used in high dosages.

Regarding the But-BA and But-BA-Lip effect in HaCaT cells, only the highest tested concentration exhibited cytotoxic effects, yet these were significantly lower compared to 5-FU employed as positive control. Since butyric acid is not toxic against skin cells and is also able to modulate cutaneous immune and inflammatory reactions [48], one may only assume that the slightly cytotoxic effect in HaCaT cells is attributable to the ester itself; however, following inclusion in liposomes, the cytotoxic effect is clearly attenuated.

The tested compounds acted as cytotoxic agents in A375 melanoma cells in a dose- and time-dependent manner; the results revealed a calculated $IC_{50}$ value of 65.9 μM for BA. While these values may seem high compared to other studies, this variability can be attributed, in part, to differences in the experimental methodologies employed, particularly regarding the concentration of DMSO used in the assays and the method of compound dilution. A similar case was reported by Suresh et al. where $IC_{50}$ values for betulinic acid, of 154 μM and 112 μM for A375 and MCF-7 cell line, respectively. [49]. The authors attribute the high $IC_{50}$ value obtained for BA on the low DMSO concentration that was used, namely 0.1% final concentration, as compared to other studies that vary this concentration from 0.5% to 2%. This low concentration led to the precipitation of BA, but the authors stated that testing BA in this suspended state mimics the in vivo scenario better in terms of drug release [49].

Out of the three tested esters, only But-BA exhibited a lower $IC_{50}$ value than BA alone, thus revealing higher cytotoxic efficacy. A similar decrease of viability in A375 melanoma cells was recorded for BA ester with myristic acid; the study demonstrated that although both BA and its ester induced cytotoxic effects, the strongest inhibition was recorded for the esterified triterpenic acid [18]. The higher $IC_{50}$ values for Pal-BA and St-BA may be correlated with the fact that when the stock solutions were diluted, at lower concentrations, the compounds formed precipitates, thus there being less available dissolved active substance at the cell site. This occurrence was previously mentioned for BA, as well, when the authors tried to obtain final BA dilution samples with a DMSO maximum concentration of 0.1% [49]. Following liposomal inclusion, an enhanced anticancer activity was recorded compared to both BA alone as well as its fatty acid esters; however, only BA-Lip, St-BA-Lip and But-BA-Lip display lower $IC_{50}$ values than the one recorded for pure BA. In agreement to the current results, numerous studies demonstrated that the inclusion of BA in surface-modified nanoformulations will result in an enhancement of their anticancer potential, as revealed in several types of cancer cells: hepatocellular carcinoma HepG2 cells ($IC_{50}$ value of 63.07 μg/mL–BA folate-functionalized liposomes) [46], cervical cancer HeLa cells (84.31% inhibition rates 48 h post-stimulation with PEGylated BA liposomes 125 μg/mL) [12], lung cancer A549 cells ($IC_{50}$ > 15 μg/mL for BA liposomes, after stimulation with a cocktail of BA, parthenolide, honokiol and ginsenoside) [45].

Although its anti-melanoma mechanisms are yet to be fully elucidated, it is known that BA acts as proapoptotic inducer in various human cancer cell lines through multiple mitochondrial-dependent mechanisms [50]; in addition, the anticancer mechanism of BA relies on the excessive production of reactive oxygen species (ROS), the regulation of the cell cycle and the inhibition of angiogenesis [51]. Apoptosis can be described as regulated cell death, typically characterized by cell shrinkage, nuclei fragmentation and dynamic membrane blebbing [52]; BA can induce cell apoptosis through several signaling pathways, being able to modulate the Bax/Bcl-2 ratio and to activate caspases-3, -7, as well as MAPK/ERK pathway [8,53]. One may assume that BA hybrid molecules such as the fatty acid esters will trigger cell death through similar apoptotic mechanisms as the parent compound; taking into consideration the effective antiproliferative activities recorded for the tested compounds, the $IC_{50}$ concentrations were selected for further examination of the

underlying molecular mechanisms of action of the hybrid compounds and their liposomal formulations as well through Hoechst (nuclei) and beta-actin (cytoskeleton) staining. Non-cancerous HaCaT cells were examined by means of the same techniques after treatment with the highest concentration previously used (100 µM) that induced cytotoxic effects. The morphological assessment in melanoma cells showed several signs of apoptosis, such as nuclei shrinkage and condensation and nuclear fragmentation, as well as disrupted cytoskeletons in melanoma cells; the proapoptotic anticancer mechanism is shared by both BA alone and the fatty acids used as esterification partners. In addition to being a source of energy and a player in the membrane structure and functions, palmitic acid exerts antitumor effects through apoptosis induction, the inhibition of tumor cell proliferation and metastasis and immunostimulation; moreover, its derivatives are able to exert additional cytoprotective effects [14]. Stearate conjugates are able to cause significant growth inhibition in cancer cells through apoptosis induction in a concentration-dependent manner while also limiting cell migration [54]. Butyric acid is among the main short-chain fatty acids secreted by the gut bacteria and may be accountable for 70% of the energy available for epithelial intestinal cells [55]; it shows strong dose-dependent anti-inflammatory and cytotoxic effects based on apoptosis induction. In particular, butyrate derivatives were revealed as antimelanoma agents acting as pro-drugs of butyric acid [56]. The delivery and release of butyric acid in cancer cell can be optimized through its inclusion in liposomes [57]. The fact that the esters synthesized in the current study adopt a proapoptotic anticancer mechanism comes as a natural consequence of the individual mechanisms of action of the conjugated molecules; also, their liposomal formulations may facilitate their delivery inside cancer cells.

Keratinocytes evaluation could not reveal any pro-apoptotic signs regardless of the tested compound. This level of anticancer selectivity is characteristic for BA alone and has long been reported starting with Pisha et al. in 1995 [58]; it is therefore to be expected that BA hybrid molecules will show similar behavior. Indeed, BA esters with palmitic and stearic acids, respectively, did not cause morphological changes in HaCaT cells which correlates well with their lack of cytotoxicity in the same cell line during cell viability studies; despite exerting slightly cytotoxic effects against HaCaT cells when used in high dosage, But-BA and But-BA-Lip did not induce any of the morphological signs associated with cell apoptosis.

Internal cellular stress initiates the mitochondrial apoptotic pathway, which involves the coordinated actions of pro-apoptotic (BAX, Bak) and anti-apoptotic (BCL-2, BCL-X, BCL-w, MCL-1, BFL-1/A1) proteins. BA was found to increase BAX expression while decreasing Bcl-2 expression, triggering apoptosis in PANC-1, SW1990, A549, HT-29, T47D, FTC 238, C6, SKNAS, TE671, Jurkat E6.1 and RPMI 8226 cells [59,60]. BA exhibited the same previously reported effect correlated with BCL2/BAX gene modulation on A375 melanoma cells, with quantitative results similar to other studies where PCR was used to determine BCL-2/BAX relative fold gene expression [60]. This pattern extends to other BA derivatives that were designed by chemical modulations of various positions (3-OH, 28-COOH or 30-allyl) [61–63] that in some cases led to a slight modification of the triterpene core as well [64]. This collectively suggests that the hydrophobic triterpene scaffold is essential for the alteration of the Bcl-2/BAX normal ratio. Considering this information, the PCR results, together with the quantitative measurements of Bax and Bcl-2 protein level, obtained for the three fatty acid esters, fell in the expected range since no significant chemical alterations were carried out on the core structure of BA. There have been a few reports in the literature related to 3-O-BA esters that show an increased pro-apoptotic activity against cancer cells when compared to BA alone by altering the expression of Bcl-2 protein family members. Saha et al. reported the biological evaluation of 3-O-dichloroacetyl-BA, which showed increased cytotoxicity and pro-apoptotic activity compared to BA by decreasing Bcl-XL levels and increasing BAX expression in MCF7 cells [65]. Drag-Zalesinska et al. showed that an increase in pro-apoptotic activity in EPP85-181P was recorded for the 3-O-lysine ester of BA, which outperformed the parent compound [66]. While this data suggests that

3-O esters of BA can have a greater pro-apoptotic effect in cancer cells than the parent compound, more research is needed to determine why But-BA was the only compound that outperformed BA.

The scratch assay is an in vitro method used to evaluate the effects of active compounds as well as their involvement in cell migration [67]. Since aberrant cell migration is a feature of cancer cells, the anti-migratory effect that would stop tumor cell invasion represents a desirable effect for any potent anticancer agent [68,69]. The antimigratory effect of our newly synthesized compounds was tested in vitro in A375 melanoma cells by using lower concentrations than cytotoxic ones; the most promising results were recorded for Pal-BA-Lip and But-BA-Lip that significantly inhibited the migration of the cancer cells in a time-dependent manner, thus revealing their potential for preventing cancer metastasis. The anti-migratory effect of BA in various cancer cells has already been described [70,71]; however, to the best of our knowledge, this is the first report on the antimigratory activity of betulinic acid esters with fatty acids which not only succeeded in inhibiting cell migration with a potency comparable to betulinic acid but in some cases even surpassing it.

Collectively, the biological data reported a significant antiproliferative and antimigratory activity for all hybrid compounds, with the butyric derivative exhibiting the strongest anticancer potential against melanoma cells. Such C-3 fatty esters of pentacyclic triterpenes also occur in plants [72] and have been under investigation for their biological effects; their natural origin indicates them as less toxic in healthy cells than synthetic drugs, a [73] fact that was verified in HaCaT keratinocytes in the current study for the similar, newly synthesized esters. Fatty acid esters may provide an additional advantage of acting as pro-drugs for the active triterpene, betulinic acid, whose release and delivery to the cancer cell can be thus influenced [74]; also, the fatty acids released as a result of ester cleavage are able to induce intrinsic cytotoxic effects through complementary pro-apoptotic activity thus adding to the overall therapeutic benefit. The inclusion of the active compounds in liposomes increased their anticancer effects in all cases, with the liposomal formulation of But-BA achieving lower $IC_{50}$ values and therefore stronger cytotoxic effects than the similar formulation of pure BA. Future research should focus on the study of the pharmacokinetic profile of these BA fatty acid prodrugs, whose delivery to the cancer cell may be controlled in order to provide prolonged anticancer activity; further in vivo studies are necessary for the complete characterization of the biological effects and toxicity of these compounds and their liposomal formulations.

## 4. Materials and Methods

### 4.1. Chemistry

#### 4.1.1. Instruments and Reagents

Betulinic acid (BA), butyryl chloride, palmitoyl chloride, stearoyl chloride, 4-dimethylaminopyrindine and and all other necessary solvents were commercially available products (Merck KGaA, Darmstadt, Germany) and were further used without any additional purification.

The 1D ([1]H and [13]C) and 2D (H,H-COSY, H,C-HSQC and H,C-HMBC) NMR experiments were performed utilizing a Bruker Avance NEO 400 MHz Spectrometer (Bruker, Billerica, MA, USA) that was equipped with a 5 mm QNP direct detection probe and z−gradients. The spectra were recorded under standard conditions in either DMSO-$d_6$ or $CDCl_3$ and were referenced on the residual peak of the solvent ([1]H: 2.51 ppm for DMSO-$d_6$ or 7.26 ppm for $CDCl_3$; [13]C: 39.5 ppm for DMSO-$d_6$ or 77.0 for $CDCl_3$).

The Biobase melting point instrument (Biobase Group in Shandong, Jinan, China), was utilized to record the melting points. Thin-layer chromatography was performed using 60 F254 silica gel-coated plates obtained from Merck KGaA in Darmstadt, Germany.

FTIR spectra were generated using KBr pellets on a Shimadzu IR Affinity-1S spectrophotometer with a 400–4000 $cm^{-1}$ range and a 4 $cm^{-1}$ resolution.

Methanolic solutions were utilized to record LC/MS spectra on an Agilent 6120 Quadrupole LC/MS system (Santa Clara, CA, USA) that was equipped with a UV detector,

an ESI ionization source, and a Zorbax Eclipse Plus C18 column (3.0 mm × 100 mm × 3.5 μm) at 40 °C in the negative ion mode. The samples were analyzed under the following conditions: 0.4 mL/min, 25 °C, and λ = 200 nm. The mobile phase was composed of a 1 mM isocratic mixture comprising 85% methanol and 15% ammonium formate. This procedure was also employed for the determination of BA content in each liposomal formulation in order to calculate the drug loading efficiency (DLE). Liposomal formulations were previously subjected to NaOH degradation and ester hydrolysis to free esterified BA. After 24 h, the aqueous solutions containing hydrolyzed BA were neutralized, extracted with ethyl acetate, evaporated and the residue was redispersed in methanol. Methanolic solutions were analyzed by the above-mentioned technique where the quantity of liposome encapsulated BA was determined using a 7-point plot calibration curve, obtained in the 50–2000 ng·mL$^{-1}$ range (R2 > 0.999 linearity). DLE for each sample was calculated as the percentage ratio between the encapsulated and total amount of used BA.

The particle morphology of the synthesized samples was analyzed in STEM Mode with a Verios G4 UC Scanning electron microscope (Thermo Scientific, Brno, Czech Republic) equipped with Energy Dispersive X-ray spectroscopy analyzer (Octane Elect Super SDD detector, Gatan, Pleasanton, CA, USA). The STEM studies were performed using the STEM 3+ detector (Bright-Field Mode) at accelerating voltage of 25 kV. For STEM analysis the samples were dispersed in water and then they were placed on carbon-coated copper grids with 300-mesh size and dried until the solvent was removed. Hydrodynamic diameters of the liposomes (BA-Lip, But-BA-Lip, Pal-BA-Lip, St-BA-Lip) were determined by DLS with a Zetasizer Pro (Malvern Panalytical, Malvern, UK). Each sample was measured at a dilution of 1:10 in deionized water and measurements were performed in triplicate. The following parameters were used for these measurements: general purpose as analysis model, automatic for size display limit mode, automatic for size threshold mode, equilibration time of 120 s and a temperature of 25 °C.

### 4.1.2. Synthesis Procedure for BA Fatty Acid Esters

A quantity of 1 mmol BA was dispersed in 20 mL of dichloromethane (DCM) and stirred for 15 min before being followed by 2 mmoles of 4-dimethylaminopyridine (DMAP). Following that, 2 mmoles of acyl chloride were added dropwise. The reaction was stirred at room temperature for 24 h. Thin-layer chromatography using chloroform as the eluent confirmed the completion of the reaction. Following water extraction, the organic phase was dried over anhydrous MgSO4 and removed by rotary evaporation. The crude product was chromatographed over silica using chloroform as the mobile phase.

*3-O-butiryl-betulinic acid* (But-BA); translucent crystals, m.p. 265–280 °C, yield 74%; $^1$H NMR (CDCl$_3$, 400 MHz, δ, ppm): 11.09 (s, 1H, COOH), 4.74 (s, 1H, H29a), 4.61 (s, 1H, H29b), 4.47 (dd, J = 4 Hz, J = 8 Hz, 1H, H3), 3.00 (m, 1H, H19), 2.29–2.26 (m, 3H, -C*H$_2$*-COO-; H15a), 2.21–0.78 (m, 42H from betulinic acid backbone, 5H from butyric acid chain). $^{13}$C NMR (CDCl$_3$, 100 MHz, δ, ppm): 182.2 (COOH), 173.6 (-COO-), 150.4 (C20), 109.7 (C29), 80.6 (C3), 56.4, 55.4, 50.4, 49.2, 46.9, 42.4, 40.7, 38.4, 38.4, 37.8, 37.1, 37.0, 36.8, 34.2, 32.1, 30.6, 29.7, 27.9, 25.4, 23.7, 20.8, 19.3, 18.6, 18.1, 16.5, 16.1, 16.0, 14.7, 13.7. FTIR [KBr] (cm$^{-1}$) relevant peaks: 2941, 2874 (C-H stretch); 1730, 1254, 1010 (ester C=O, C-C-O, O-C-C stretch); ESI-MS, *m/z* = 525 [M-H$^+$]$^-$.

*3-O-palmitoyl-betulinic acid* (Pal-BA); white powder, m.p. 160–170 °C, yield 70%; $^1$H NMR (CDCl$_3$, 400 MHz, δ, ppm): 11.15 (s, 1H, COOH), 4.74 (s, 1H, H29a), 4.61 (s, 1H, H29b), 4.47 (dd, J = 4 Hz, J = 8 Hz, 1H, H3), 3.00 (m, 1H, H19), 2.30–2.27 (m, 3H, -C*H$_2$*-COO-; H15a), 2.21–0.86 (m, 42H from betulinic acid backbone, 29H from palmitic acid chain). $^{13}$C NMR (CDCl$_3$, 100 MHz, δ, ppm): 182.3 (COOH), 173.8 (-COO-), 150.4 (C20), 109.7 (C29), 80.6 (C3), 56.4, 55.4, 50.4, 49.2, 46.9, 42.4, 40.7, 38.4, 38.4, 37.8, 37.1, 37.0, 34.8, 34.2, 32.1, 31.9, 30.6, 29.7, 29.6, 29.6, 29.5, 29.4, 29.2, 29.2, 28.0, 25.4, 25.2, 23.7, 22.7, 20.8, 19.3, 18.1, 16.5, 16.1, 16.0, 14.7, 13.7. FTIR [KBr] (cm$^{-1}$) relevant peaks: 2930, 2859 (C-H stretch); 1730, 1242, 1010 (ester C=O, C-C-O, O-C-C stretch); ESI-MS, *m/z* = 693 [M-H$^+$]$^-$.

*3-O-stearoyl-betulinic acid* (St-BA); white powder, m.p. 150–162 °C, yield 65%; $^1$H NMR (CDCl$_3$, 400 MHz, δ, ppm): 10.85 (s, 1H, COOH), 4.74 (s, 1H, H29a), 4.61 (s, 1H, H29b), 4.47 (dd, J = 4 Hz, J = 8 Hz, 1H, H3), 3.00 (m, 1H, H19), 2.30–2.27 (m, 3H, -CH$_2$-COO-; H15a), 2.21–0.88 (m, 41H from betulinic acid backbone, 31H from stearic acid chain). $^{13}$C NMR (CDCl$_3$, 100 MHz, δ, ppm): 182.0 (COOH), 173.8 (-COO-), 150.4 (C20), 109.7 (C29), 80.6 (C3), 56.4, 55.4, 50.4, 49.3, 46.9, 42.4, 40.7, 38.4, 38.4, 37.8, 37.1, 37.1, 34.9, 34.2, 32.2, 31.9, 30.6, 29.7, 29.6, 29.6, 29.5, 29.4, 29.3, 29.2, 28.0, 25.4, 25.2, 23.7, 22.7, 20.8, 19.3, 18.2, 16.5, 16.2, 16.0, 14.7, 13.7. FTIR [KBr] (cm$^{-1}$) relevant peaks: 2924, 2853 (C-H stretch); 1730, 1242, 1010 (ester C=O, C-C-O, O-C-C stretch); ESI-MS, $m/z$ = 721 [M-H$^+$]$^-$.

### 4.1.3. Synthesis Procedure for BA Fatty Acid Esters Liposomal Formulations

The liposomes were made using the thin-layer hydration method using a slightly modified, previously published procedure [28]. A quantity of 100 mg of L-α- phosphatidylcholine followed by 0.03 mmols of cholesterol, 0.004 mmols of DSPE-PEG2000, and 0.02 mmols of triterpene were dissolved in chloroform and stirred until a clear solution was obtained. The solvent was then removed with a rotary evaporator, and the resulting lipid film was hydrated with 10 mL of phosphate buffer saline (PBS). The mixture was allowed to hydrate for 24 h before being redispersed for 30 min using sonication. To remove unencapsulated BA, the emulsion was centrifuged at 3000 rpm for 10 min, 3 times, the supernatant was collected and was stored at 4 °C.

### 4.2. In Vitro Assessment

### 4.2.1. Cell Culture

The cell lines selected for our study were noncancerous human keratinocytes—HaCaT, acquired from CLS Cell Lines Service GmbH (Eppelheim, Germany), and human malignant melanoma—A375, purchased from American Type Culture Collection (ATTC, Lomianki, Poland). The cells were acquired as frozen items and were stored in liquid nitrogen. Both HaCaT and A375 were cultured and propagated in Dulbecco's Modified Eagle Medium (DMEM) high glucose, supplemented with 10% fetal bovine serum (FBS) and 1% antibiotic mixture of Penicillin/Streptomycin (100 IU/mL). The cells were maintained in a humified incubator with 5% CO$_2$ at 37 °C.

### 4.2.2. Cell Viability Assessment

Alamar blue assay. The Alamar blue staining method was used to determine the cell viability of HaCaT and A375 cells, post stimulation with increasing concentrations (10, 25, 50, 75 and 100 μM) of BA and its fatty esters Pal-BA, St-BA and But-BA, and the PEGylated liposomes Pal-BA-Lip, St-BA-Lip, But-BA-Lip, BA-Lip, the liposome in a free form (Lip) and using 5-fluorouracil as positive control, for 24 h and 48 h. The tested concentrations for the liposomal formulations were obtained considering previously recorded DLE values. The cells (1 × 10$^4$ cells/well) were seeded onto 96-well plates and incubated 37 °C and 5% CO$_2$ until reaching 80–85% confluence. The cell number was determined with Trypan blue coloring using an automated cell counting device (Thermo Fisher Scientific, Inc., Waltham, MA, USA). The used medium was removed using an aspiration station and replaced with fresh medium containing the tested compounds. The tested concentrations (10, 25, 50, 75 and 100 μM) were prepared using stock solutions of 20 mM and the final concentration of DMSO did not exceed 0.5%. After 24 h, and 48 h, respectively, the cells were stained with Alamar blue 0.01% by adding to each well 20 μL Alamar blue 0.01%, obtaining a finale volume of 220 μL/well and then incubated for another 3 h in a humified incubator with 5% CO$_2$ at 37 °C. The absorbance was measured at two wavelengths, 570 nm, and 600 nm using xMark™ Microplate Spectrophotometer, Bio-Rad (Hercules, CA, USA).

### 4.2.3. Fatty Ester Derivatives Effects on Cell Morphology

The A375 melanoma cells and the human keratinocytes HaCaT cells were seeded onto 12-well plates at initial density of 2 × 10$^5$ cells/well until reaching 80–85% confluence.

Afterwards, the cells were stimulated with the tested compounds for 48 h at the highest tested concentration (100 μM) for HaCaT cells and the corresponding $IC_{50}$ values for A375. All the cells were stimulated with 5-FU as positive control. After 48 h, the morphology of the cells was evaluated using the EVOS™ M5000 Imaging System equipped with a highly sensitive CMOS camera (Thermo Fisher Scientific, Inc., Waltham, MA, USA).

### 4.2.4. Immunofluorescence Assay—Morphological Assessment of Apoptotic Cells

Hoechst staining was used to assess the nuclear localization and determine signs of apoptosis (fragmentation, shrinkage), while beta-actin staining was utilized to determine the cytoplasmatic localization. HaCaT and A375 cells were seeded onto 12-well plates at initial density of $2 \times 10^5$ cells/well. After reaching 80–85% confluence, the cells were stimulated for 48 h with the tested compounds at their $IC_{50}$ values for A375 cells and at 100 μM—the highest concentration for HaCaT cells. Separately, some wells were treated with staurosporine 5 μM as positive control. After 48 h the old medium was removed and the cells were fixed with methanol for 15 min, permeabilized with Triton X 0.01% in PBS for 15 min and blocked with 3% BSA for 30 min at room temperature. Later on, the cells were stained with beta actin monoclonal antibody using a 1:2000 dilution (Thermo Fisher Scientific, Inc., Waltham, MA, USA) in BSA 3% for 1 h at room temperature and then were incubated with Alexa fluor 488 goat-anti mouse secondary antibody (Thermo Fisher Scientific, Inc., Waltham, MA, USA) at a dilution of 1:5000 in BSA 3% for 30 min in the dark, at room temperature. Finally, for nuclear staining the Hoechst 33258 solution was added for 5 min. The nuclear and cytoplasmatic modifications were analyzed using the EVOS™ M5000 Imaging System equipped with a highly sensitive CMOS camera (Thermo Fisher Scientific, Inc., Waltham, MA, USA).

### 4.2.5. Real-Time PCR Quantification of Apoptotic Markers

The total RNA content was extracted using the peqGold RNAPureTM Package (Peqlab Biotechnology GmbH, Erlangen, Germany) following the manufacturer's instructions, and the total concentration of RNA was measured using a DS-11 spectrophotometer (DeNovix, Wilmington, DE, USA). The Maxima® First Strand cDNA Synthesis Kit (Thermo Fisher Scientific, Inc., Waltham, MA, USA) was used for reverse transcription, and the samples were incubated in the Tadvanced Biometra Product line (Analytik Jena AG, Göttingen, Germany) using the following thermal cycle: 10 min at 25 °C, 15 min at 50 °C, and 5 min at 85 °C. Quantitative real-time PCR was conducted employing a Quant Studio 5 real-time PCR system (Thermo Fisher Scientific, Inc., Waltham, MA, USA). The analysis was performed using 20 μL aliquots containing Power SYBR-Green PCR Master Mix (Thermo Fisher Scientific, Inc., Waltham, MA, USA), sample cDNA, the sense and antisense primer and pure water. The primer pairs used for this method included 18S, used as housekeeping gene (sense: 5′ GTAACCCGTTGAACCCCATT 3′; antisense: 5′ CCATC-CAATCGGTAGTAGCG 3′), Bax (sense: 5′ GCCGGGTTGTCGCCCTTTT 3′; antisense: 5′CCGCTCCCGGAGGAAGTCCA 3′) and Bcl-2 (sense: 5′CGGGAGATGTCGCCCCTGGT 3′; antisense: 5′GCATGCTGGGGCCGTACAGT 3′) (Thermo Fisher Scientific, Inc., Waltham, MA, USA). Normalized, relative expression results were calculated using the comparative threshold cycle method ($2^{-\Delta\Delta Ct}$).

### 4.2.6. Scratch Assay

The scratch test was performed in order to assess the regressive effect of Pal-BA, St-BA, But-BA, Pal-BA-Lip, St-BA-Lip, But-BA-Lip, compared to the parent compound BA and BA-Lip on the invasion capacity of malignant melanoma A375 cells. The cells ($2 \times 10^5$/well) were seeded onto 12-well plates until reaching 80–85% confluence. Then, the old medium was removed and the attached cells were scratched onto the diameter of the well using a sterile pipette tip. After washing the cells with warm PBS, the cells were stimulated with each tested compound at 10 μM. To establish the growing rate of the stimulated cells compared to control, the wells were photographed at 0, 24 and 48 h using the Olympus

IX73 inverted microscope (Olympus, Tokyo, Japan). The cells Sense Dimension software (version 1.8) was utilized for analyzing cell migration for each cell line.

The following formula was used in order to calculate the scratch closure rate [75]:

$$\text{Scratch closure rate} = \left[ \frac{At_0 - At}{At_0} \right] \times 100$$

where $At_0$ is the scratch at time 0 h and $At$ is the scratch area at 24 h or 48 h.

### 4.3. BAX and Bcl-2 Detection

The apoptotic protein markers, Bcl-2 and BAX, were determined in A375 melanoma cells 48 h post-treatment with BA fatty esters (Pal-BA, St-BA, But-BA) and their liposomes (Pal-BA-Lip, St-BA-Lip, But-BA-Lip) as well as with the parent compound BA and its liposome BA-Lip using their respective IC50 values. Bcl-2 (ab119506) and BAX (ab199080) concentrations in cell lysates were assessed using the Elisa kits purchased from Abcam, according to the manufacturers' protocols [76]. To conduct the assay, samples or standards are added to the wells, followed by the antibody mix. After incubation, unbound material is washed away. TMB substrate is added, and in the presence of HRP, it catalyzes a reaction producing a blue color. This color reaction is halted by adding Stop Solution, resulting in a color change from blue to yellow. Optical densities were read using a microplate reader (xMark™ Microplate Spectrophotometer, Biorad, Hercules, CA, USA) at 450 nm.

### 4.4. Statistical Analysis

The statistical tests were carried out using one-way ANOVA followed by Dunnett's post-test (GraphPad Prism version 6.0.0, GraphPad Software, San Diego, CA, USA). The differences between the groups were considered statistically significant if $p < 0.05$, as follows: * $p < 0.05$, ** $p < 0.01$ and *** $p < 0.001$. The $IC_{50}$ values presented in Table 3 were calculated using GraphPad Prism version 6.0.0 (GraphPad Software, San Diego, CA, USA).

## 5. Conclusions

Our study described the synthesis and cytotoxic evaluation of a series of BA derivatives obtained through the esterification with fatty acids such as palmitic, stearic and butyric acid, as well as their surface-modified liposomal nanoformulations. The synthesis protocol elicited good yields for each BA fatty ester, which were further incorporated in PEGylated liposomes. Both fatty esters as well as their liposomal nanoformulations exhibited significant cytotoxic effects in A375 human melanoma cells, comparable and, for some compounds, stronger than those recorded for parent compound BA and its liposomal formulation, BA-Lip. A related cytotoxic corelated pro-apoptotic effect was observed for all compounds and their subsequent formulations. Similar to the cytotoxicity results, the tested derivatives and formulations decreased the expression of the antiapoptotic marker Bcl-2 and increased the expression of the proapoptotic marker BAX. The cytotoxicity assessment against HaCaT cells revealed only slight cytotoxic effects for But-BA and its respective liposomal formulation, such effects lacking for palmitic and stearic BA conjugates. In all cases, the inclusion in liposomes enhanced the anticancer potential of the active compound. Our findings suggest that the further optimization of these compounds, particularly focusing on improving their solubility and formulation dynamics, could lead to significant advancements in cancer therapy. Future studies should prioritize pharmacokinetic evaluations and in vivo assessments to better understand the therapeutic potential of these compounds. Ultimately, our study provides new avenues for effective cancer treatment strategies, describing the successful development of BA derivatives and their liposomal formulations that may offer novel additions to the development of similar compounds with increased and selective cytotoxic effect.

**Supplementary Materials:** The following supporting information can be downloaded at: https://www.mdpi.com/article/10.3390/pr12020416/s1, Figure S1. 1H NMR spectrum of 3-O-butyryl-betulinic acid

(But-BA); Figure S2. 13C NMR spectrum of 3-O-butyryl-betulinic acid (But-BA); Figure S3. 13C DEPT NMR spectrum of 3-O-butyryl-betulinic acid (But-BA); Figure S4. H,H-COSY NMR spectrum of of 3-O-butyryl-betulinic acid (But-BA); Figure S5. H,C-HSQC NMR spectrum of 3-O-butyryl-betulinic acid (But-BA); Figure S6. H,C-HMBC NMR spectrum of 3-O-butyryl-betulinic acid (But-BA); Figure S7. 1H NMR spectrum of 3-O-palmitoyl-betulinic acid (Pal-BA); Figure S8. 13C NMR spectrum of 3-O-palmitoyl-betulinic acid (Pal-BA); Figure S9. 13C DEPT NMR spectrum of 3-O-palmitoyl-betulinic acid (Pal-BA); Figure S10. H,H-COSY NMR spectrum of 3-O-palmitoyl-betulinic acid (Pal-BA); Figure S11. H,C-HSQC NMR spectrum of 3-O-palmitoyl-betulinic acid (Pal-BA); Figure S12. H,C-HMBC NMR spectrum of 3-O-palmitoyl-betulinic acid (Pal-BA); Figure S13. 1H NMR spectrum of 3-O-stearoyl-betulinic acid (St-BA); Figure S14. 13C NMR spectrum of 3-O-stearoyl-betulinic acid (St-BA); Figure S15. 13C DEPT NMR spectrum of 3-O-stearoyl-betulinic acid (St-BA); Figure S16. H,H-COSY NMR spectrum of 3-O-stearoyl-betulinic acid (St-BA); Figure S17. H,C-HSQC NMR spectrum of 3-O-stearoyl-betulinic acid (St-BA); Figure S18. H,C-HMBC NMR spectrum of 3-O-stearoyl-betulinic acid (St-BA); Figure S19. FTIR spectra of BA, But-BA, Pal-BA and St-BA; Figure S20. The evaluation of morphological changes of HaCaT cells 48 h treatment with BA, BA-Lip, 5-FU (A), Pal-BA, Pal-BA-Lip, St-BA (B), St-BA-Lip, But-BA and But-BA-Lip (C); Figure S21. The evaluation of morphological changes of A375 cells after 0 h and 48 h treatment with BA, BA-Lip, 5-FU (A), Pal-BA, Pal-BA-Lip, St-BA (B), St-BA-Lip, But-BA and But-BA-Lip (C).

**Author Contributions:** Conceptualization, A.M. (Andreea Milan), M.M. and C.Ș.; methodology, A.M. (Andreea Milan), M.M., A.M. (Alexandra Mioc), M.B.-P., R.R., G.M., S.R., N.M. and C.Ș.; formal analysis, M.B.-P., G.M. and S.R.; validation, A.M. (Andreea Milan), A.M. (Alexandra Mioc) and M.M.; investigation, A.M. (Andreea Milan), A.M. (Alexandra Mioc) and M.M.; software, A.M. (Andreea Milan), A.M. (Alexandra Mioc), M.B.-P., R.R., S.R. and I.Ş.; writing—original draft preparation, A.M. (Andreea Milan); writing—review and editing A.M. (Andreea Milan), A.M. (Alexandra Mioc), M.M. and C.Ș.; visualization, R.R., I.Ş. and C.Ș.; supervision, A.M. (Alexandra Mioc), M.M. and C.Ș.; project administration, M.M. and C.Ș.; funding acquisition, M.M. All authors have read and agreed to the published version of the manuscript.

**Funding:** This research was funded by the University of Medicine and Pharmacy "Victor Babes" Timisoara, grant number 26679/09.11.2022 (M.M.).

**Data Availability Statement:** The original contributions presented in the study are included in the article/Supplementary Material, further inquiries can be directed to the corresponding author/s.

**Conflicts of Interest:** The authors declare no conflicts of interest.

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
