# Peer review of "Exploring the Antimelanoma Potential of Betulinic Acid Esters and Their Liposomal Nanoformulations"

_processes, doi:10.3390/pr12020416_

Round 1

Reviewer 1 Report

Comments and Suggestions for Authors

Brief Summary

The manuscript entitled “Exploring the Antimelanoma potential of Betulinic Acid Esters and their Liposomal Nanoformulations” was submitted to the Special Issue of Processes – Natural Compounds Applications in Drug Discovery and Development. It focuses on melanoma, a highly aggressive form of skin cancer that is in need for more effective therapeutic alternatives. The plant-derived betulinic acid (BA) has been researched for its anticancer activity. Furthermore, the association of pharmacologically active natural compounds to drug delivery systems, namely liposomes, has been a widely employed strategy to improve their therapeutic index. These lipid-based nanosystems are well established in the clinic, demonstrating their success as therapeutic tools against several human diseases, including cancer. In this work, the authors synthetized BA derivatives to improve cellular uptake and pharmacological effect. Furthermore, the incorporation in liposomes was proposed as an effective approach to maximize the antimelanoma activity of BA derivatives.  

General concept comments

  • Is the manuscript clear, relevant for the field and presented in a well-structured manner? 

The manuscript is clear, although some grammar revisions may be required. Overall, it is relevant, providing information on the mechanisms of action of novel BA derivatives and the successful use of liposomes as tools to enhance the antitumor activity.

  • Are the cited references mostly recent publications (within the last 5 years) and relevant? Does it include an excessive number of self-citations?

The cited references are mostly recent and relevant. The self-citations are not excessive and are relevant to the manuscript.

  • Is the manuscript scientifically sound and is the experimental design appropriate to test the hypothesis?

In general, the manuscript is scientifically sound, with a strong basis on previous data on BA and other BA derivatives. However, there are some important data that should be included, such as the physicochemical properties of BA derivatives and further stability studies of liposomal formulations.

  • Are the manuscript’s results reproducible based on the details given in the methods section?

Overall, sufficient detail is given; however, more detailing and/or correction is required in some sections of the methods.

  • Are the figures/tables/images/schemes appropriate? Do they properly show the data? Are they easy to interpret and understand? Is the data interpreted appropriately and consistently throughout the manuscript? Please include details regarding the statistical analysis or data acquired from specific databases.

Most Figures and Tables require major corrections (e.g. Figure 13) or a replacement (e.g. Figure 12). Some Figures are not very clear and difficult to understand. For instance, in Figure 10, arrows indicate some cells with potential signs of apoptosis; however, the images do not allow to easily see these signs. Also, in Figure 13, it is difficult to understand the wound limits. The statistical analysis described in the Methods section/Figure 12 needs to be corrected.

  • Are the conclusions consistent with the evidence and arguments presented?

The conclusions are consistent with the presented data and discussion.

  • Please evaluate the ethics statements and data availability statements to ensure they are adequate.

The ethics statements are adequate. The data availability statement is missing.

Specific comments

A.      In paragraph 55-57, the authors describe BA’s mechanisms of action. However, reference 8 addresses only asiatic acid, another pentacyclic triterpenoid. Please correct accordingly.

B.      In lines 60-61, the authors refer to liposomal nanoformulations. However, reference 12 reports the synthesis of silver nanoparticles using rosemary leaves extract. Please correct accordingly.

C.       In line 68, please replace “G365” with “G361”, as G365 cell line was not the one used by the referenced work.

D.      In lines 69-71, please add a reference to the paragraph “Chodurek et al. have tested its 69 chemopreventive effect against A375 melanoma cells while also revealing that sodium 70 butyrate was able to inhibit cell proliferation.”

E.       In the results section:

a.       In Figure 1 caption, please define the DCM and DMAP abbreviations.

b.       The authors should include a table summarizing the physicochemical properties of each newly synthetized BA derivative.

c.       The authors have assessed the liposomes’ stability over one week, in terms of mean size and PI. Have the authors determined the stability in terms of loaded compounds after one week of storage, as well as plasma stability? Since the chemical modification with esters is used to improve lipophilicity, then the passive membrane permeability is also enhanced, including through the liposomes’ bilayer. Furthermore, the zeta potential (mV) of developed nanoformulations should be assessed to determine if the presence of compounds within the lipid bilayer may affect the surface charge of liposomes. Please comment.

d.       In Figure 2, please increase the visibility of the scale bar in each picture. Also, please confirm in the caption if the scale bar for (E) is 100 or 200 nm.

e.       In Figure 2C, the liposomes display an irregular surface, compared to the other imaged liposomes. Do the authors have a possible explanation for this fact?

f.        In line 169, please define the abbreviation of 5-FU.

g.       In Figures 5, 6 and 11, please define if the statistical significance is versus the control cells.

h.       In Figure 5, * p < 0.05 is not represented in the graphs, and in Figure 6 the same applies to * p < 0.05 and ** p < 0.01. Please also check this in Figure 11.

i.         In the text and in Table 1, please indicate the incubation time (24 or 48 h) that the values of IC50 correspond to. Also, the viability experiments were performed independently three times. The authors should include the SD for the IC50 values in Table 1.

j.         In Section 2.3., the authors should also report on 5-FU group.

k.       I would suggest Figures 7 and 8 to be included in the Supplementary Material.

l.         In Figures 9 and 10, I suggest the authors include what images correspond to the individual staining (nuclei and beta-actin staining) and the merged picture. In these same figures, please include in the caption the abbreviation for Staurosporine (STZ).

m.     In Figure 10, the authors indicate signs of apoptosis with yellow arrows and describe these signs in the text. Nevertheless, I find it difficult to perceive from the images provided. If the authors have images of some of these apoptotic cells in a higher magnification, I believe it would benefit the readers understanding.

n.       In Figure 12, please correct the graphs using values obtained with the appropriate Scratch Closure Rate formula. In line 673, replace the incorrect equation [(𝐴𝑡0−𝐴𝑡)/𝐴𝑡]×100 by [(𝐴𝑡0−𝐴𝑡)/𝐴0]×100.

o.       Please pay special attention to Figure 13. BA T48 and BA-Lip T48 have the same image. Please replace them with the correct ones. The same applies to But-BA and But-BA-Lip.

F.       In the Discussion section:

a.       In lines 311-312, the authors wrote “…natural compounds have the ability to trigger multiple oncogenic signaling pathways…”. I would suggest the authors replace “trigger” with a more appropriate term (e.g.: affect/modulate/influence), otherwise the sentence is incorrect.

b.       In lines 313-314, in the sentence “Betulinic acid which can be found in many species…” the authors should include “tree and plant species”, as mentioned in the Introduction section.

c.       In lines 482-483, the sentence “…butyric derivative exhibiting the strongest anticancer potential accompanied by lower cytotoxic effects in healthy cells” should be revised since, among all developed formulations,  But-BA and But-BA-Lip induced cytotoxicity in HaCaT cells.

G.      In Materials and Methods:

a.       In section 4.1.1, line 539, the authors refer to “Lip-Gal”, a term that only appears once in the manuscript. Please correct accordingly.

b.       In section 4.1.3., the passing through a 0.2 μm polycarbonate membrane only serves to reduce the mean size. Both the liposomes and the extraliposomal medium (possibly containing non-incorporated compounds) will cross the membrane. This, in turn, greatly affects the drug loading calculations. There are appropriate methods to separate non-incorporated lipophilic compounds, namely gel filtration. Have the authors considered these options? Another important piece of information to include is how the lipid content was determined.

c.       In section 4.2.1, in lines 605-608, I would suggest removing the paragraph “After reaching 80-90% confluence cells were stimulated with the tested compounds (10, 25, 50, 75 and 100 μΜ) for 24h and 48h, respectively. The cell number was determined with Trypan blue colouring using an automated cell counting device (Thermo Fisher Scientific, Inc., Waltham, MA, USA).” This information should be placed in the next section 4.2.2 when describing the seeding of the cells in 96-well plates.

d.       In section 4.2.2, line 613, the authors included “free BA” in the listing of PEGylated liposomes. Please correct accordingly.

e.       In line 617, the authors wrote that after seeding in 96-well plates, cells were exposed to tested formulations when reaching 80-85% confluence. With this high confluence at the start of incubation, the Control cells, especially A375 cells, would be overconfluent after 48 h. Did the authors notice any changes in cells’ monolayer after 48 h?

f.        In lines 618-620, the authors should specify that this refers to the compounds in the free form.

g.       In lines 620-621, please detail the volume/well of Alamar blue solution and the procedure before adding the staining.

h.       In section 4.2.5, line 664, please include if the cell number is per well or per mL. Also, does the Sense Dimension software, when analyzing the images, generate any kind of visible limits/lines? It would be quite beneficial to include these analyzed images instead of the current Figure 13) that is somewhat difficult to interpret.

i.         In line 668, the authors wrote “…tested compound using their IC50 values.” Nevertheless, in line 281, the tested concentration indicated is 10 μM for all formulations.

j.         In line 673, please correct the Scratch Closure Rate formula.

k.       In section 4.3., the statistical analysis referred is Dunnett’s. However, in Figure 12, the authors indicate the Bonferroni’s Comparison Test. Please correct accordingly. In Figures 5 and 6 the statistical information is missing. In this section, the authors refer that the IC50 values were calculated using GraphPad Prism and Microsoft Excel. Were the values similar or did they differ? In Table 1, the values correspond to which program?

l.         A section in Methods describing the procedures for the results of bright field “Fatty ester derivatives effects on cell morphology” (Figures 7 and 8) is needed.

H.      In References section:

a.       References 27 and 74 are the same. Please correct accordingly.

b.       References 14 and 52 are the same. Please correct accordingly.

Rating the Manuscript

  • Novelty: Is the question original and well-defined? Do the results provide an advancement of the current knowledge?

Although the tested compounds were newly synthetized, the approach itself is not considered novelty. The authors clearly define their proposed strategy and the obtained results provide novel data on these new formulations as potential antimelanoma agents.

  • Scope: Does the work fit the journal scope*?

Yes, the work fits the journal scope. The manuscript is within the scope of Pharmaceutical Processes, namely pharmaceutical synthesis, nanomedicine and nanotechnology, drug delivery systems and mechanisms of drugs.

  • Significance: Are the results interpreted appropriately? Are they significant? Are all conclusions justified and supported by the results? Are hypotheses carefully identified as such?

The results are correctly interpreted and significant since these are newly synthetized BA derivatives, as well as the corresponding liposomal formulations. The conclusions correlate with the reported data and the authors clearly define their purpose.

  • Quality: Is the article written in an appropriate way? Are the data and analyses presented appropriately? Are the highest standards for presentation of the results used?

Some minor grammar revisions are required. The presentation of some data and analysis needs revision.

  • Scientific Soundness: Is the study correctly designed and technically sound? Are the analyses performed with the highest technical standards? Is the data robust enough to draw conclusions? Are the methods, tools, software, and reagents described with sufficient details to allow another researcher to reproduce the results? Is the raw data available and correct (where applicable)?

Overall, the study is correctly designed, despite lacking some technical soundness. Some data analysis should be revised (as detailed below). The obtained data are robust enough to draw the presented conclusions. The Methods need to be revised in order to provide sufficient and correct information to reproduce the results.

  • Interest to the Readers: Are the conclusions interesting for the readership of the journal? Will the paper attract a wide readership, or be of interest only to a limited number of people? (Please see the Aims and Scope of the journal.)

The manuscript focuses on the synthesis of new derivatives from a natural compound and the subsequent association to a drug delivery system. Here, the authors also assessed the in vitro antimelanoma potential of developed formulations, conducting different assays to define the mechanism(s) of action. As this manuscript encompasses pharmaceutical synthesis, nanomedicine and nanotechnology, as well as mechanisms of drugs, it may interest a wide readership.

  • Overall Merit: Is there an overall benefit to publishing this work? Does the work advance the current knowledge? Do the authors address an important long-standing question with smart experiments? Do the authors present a negative result of a valid scientific hypothesis?

Overall, this work provides novel information regarding new synthetized derivatives of betulinic acid, advancing the knowledge in this the pharmaceutical synthesis area, as well as in the application of liposomes for their delivery. In terms of experiments, the authors resorted to several assays to determine the mechanism(s) of action of the developed formulations. These assays provided the necessary data for the presented work; however, the approach is not new.

  • English Level: Is the English language appropriate and understandable?

The English language is mostly appropriate and understandable, requiring only minor grammar revisions.

Comments on the Quality of English Language

The English quality is adequate, although some minor revisions are needed.

Reviewer 2 Report

Comments and Suggestions for Authors

Presented manuscript describes the synthesis of several esters of betulinic acid with fatty acids and butyric acid. The compounds were also used for liposome formulations and tested for their cytotoxic activity on melanoma cancer cells and immortalized keratinocytes as control. Although the work seem interesting to me, in my opinion there are several issues that need more concern:

1. The cytotoxicity of both, the esters and their liposomic formulation is around 60 uM which is very low, in many studies, 50uM is the highest concentrations at which the compounds are tested and if the IC50 value is higher, the compounds are considered inactive.

2. I am sure that all of the esters (maybe with the exception of the butyric ester but likely not) were not dissolved at 60 uM in water-based media with only 0.5% DMSO. In my opinion, all of the compounds were either precipitates or gel-kind of clusters whuch may be the reason of the low activity, the accessibility of such precipitates is much lower. Liposomes probably do not have this issue but their activity was as low as the bare esters.

3. The authors show the increase of the expression of BAX protein in compound-treated cells and decrease of anti-apoptotic protein Bcl-2 at mRNA level. This seems legit and in concordance with the majority published cytotoxicity studies of BA derivatives. It would be great if the authors were able to show this at the protein level using Western blot, this shows this mechanism unabiquituoisly.

4. The article is completely missing any conclusion ,what is this useful and in which way and where to go further? In case of compounds of IC50 at around 60 uM, it is obvious that some significant improvement must be achieved in order to make the compounds potentially useful in cancer treatment.

To conclude, although the activities are rather low and it is not very likely to use these compounds in the treatment, I believe that the authors brought some new ideas to the topic, it may be generally useful for triterpenes to use them in a form of liposomes and there is a room for improvement. Therefore I reccomend this article to be published after some major adjustments are done.

Comments on the Quality of English Language

Language is OK, the article is well readable, logic etc. 

Round 2

Reviewer 1 Report

Comments and Suggestions for Authors

The authors have addressed the reviewers' questions and performed major appropriate corrections to the manuscript.

Some comments:

a) In Figure 13A, the authors indicate 5-FU; however, this is not represented in the images that correspond to Control, BA and BA-Lip.

b) Also, images BA-T48 and BA-Lip T48 are duplicates. They only differ on the field of view, higher or lower, of the same image. The correction of the images has already been asked in the first review report. this is inappropriate and may jeopardize scientific soundness. I understand that mistakes can be made so, please, correct the figures and carefully confirm the data presented.

Author Response

The authors have addressed the reviewers' questions and performed major appropriate corrections to the manuscript.

Some comments:

  1. a) In Figure 13A, the authors indicate 5-FU; however, this is not represented in the images that correspond to Control, BA and BA-Lip.

Response: We apologize for this mistake. We performed the necessary correction.

  1. b) Also, images BA-T48 and BA-Lip T48 are duplicates. They only differ on the field of view, higher or lower, of the same image. The correction of the images has already been asked in the first review report. this is inappropriate and may jeopardize scientific soundness. I understand that mistakes can be made so, please, correct the figures and carefully confirm the data presented.

Response: We apologize for the oversight on our part and truly appreciate your patience and understanding throughout this process. We performed the necessary correction.

Once again, thank you for your time. Your expertise and guidance have undoubtedly contributed to enhancing the quality and accuracy of our paper.

Reviewer 2 Report

Comments and Suggestions for Authors

The changes improved the manuscript in a manner that it may be accepted for publication. I believe, this will be interesting for the readers of this journal.

Comments on the Quality of English Language

Language is OK

Author Response

We would like to thank you for your invaluable feedback, suggestions and recommendations that led to significant improvements in the presentation of our article.